# Pinpointing Attention-Causal Communication in Language Models

**Gabriel Franco**
Department of Computer Science
Boston University
gvfranco@bu.edu

**Mark Crovella**
Department of Computer Science and
Faculty of Computing & Data Sciences
Boston University
crovella@bu.edu

## Abstract

The attention mechanism plays a central role in the computations performed by transformer-based models, and understanding the reasons why heads attend to specific tokens can aid in interpretability of language models. Although considerable work has shown that models construct low-dimensional feature representations, little work has explicitly tied low-dimensional features to the attention mechanism itself. In this paper we work to bridge this gap by presenting methods for identifying *attention-causal communication*, meaning low-dimensional features that are written into and read from tokens, and that have a provable causal relationship to attention patterns. The starting point for our method is prior work [1–3] showing that model components make use of low dimensional communication channels that can be exposed by the singular vectors of QK matrices. Our contribution is to provide a rigorous and principled approach to finding those channels and isolating the attention-causal signals they contain. We show that by identifying those signals, we can perform prompt-specific circuit discovery in a single forward pass. Further, we show that signals can uncover unexplored mechanisms at work in the model, including a surprising degree of global coordination across attention heads.

## 1 Introduction

Transformer-based language models exhibit remarkable abilities [4–6] and consequently have been very widely deployed. However, it is quite challenging to explain *how* language models are able to accomplish such sophisticated tasks [7]. Nonetheless, progress on interpretability for language models is critical, e.g., to build a foundation for improving model safety and alignment [8].

A meaningful explanation for model behavior should capture causal, rather than merely correlative, relationships. Elucidating causal relationships within language models has been largely approached using interventions during model execution [9–15], a method termed causal mediation analysis [16]. However, in this paper we treat the model itself as a structural causal model [13, 17] and thereby exactly identify the impact of a given counterfactual on the model's computation.

We focus on causal analysis of a key model computation: *attention*. In the attention mechanism a head uses its QK matrix to compute an attention weight for each token pair, and when the weight is large, we say the head *attends* to the token pair, resulting in a significant movement of information within the model. The attention mechanism has been celebrated for its ability to respond to precise features in activations and enable sophisticated behaviors [18–20]. From the standpoint of causal analysis, the attention mechanism raises a basic question: when a head attends to a token pair, what is the most useful counterfactual? That is, what are the most informative features present in the tokens that *explain* the head's attention to the token pair [16]? Answering this question has potential

39th Conference on Neural Information Processing Systems (NeurIPS 2025).

to make progress on critical challenges in interpretability. As shown in [21], model components work together to implement complex behaviors by reading and writing to residuals, so precisely identifying attention-causal features uncovers key communication taking place within the model.

Leverage in addressing our question comes from the large amount of work showing that many important features in models are encoded in one-dimensional subspaces [15, 22–28] or in low-dimensional subspaces [29–32]. Hence we seek to explain a head's attention in terms of the presence or absence of features encoded in low-dimensional subspaces of the residuals.

This suggests a focus on parsimony in selecting counterfactuals. We frame the question as follows: Given an attention head attending to a pair of tokens, for each token we seek to identify a low-dimensional subspace and a small number of upstream components, such that if the components had not written into the token in that subspace, the head would not have attended to the token pair. We show that this strategy identifies low-dimensional *signals* that are causal for attention, directly provide efficient and useful circuit traces, and can uncover model-wide control mechanisms at work.

Our approach to this question starts from prior work [1–3] showing that the singular vectors of QK matrices will tend to be aligned with important features in residuals. Hence we start by decomposing residuals in the spectral bases of QK matrices. However, placing activations in a new basis is not enough. Model heads use the $\mathrm{Softmax}$ function to compute attention; this nonlinearity complicates the question of deciding which components of a residual are causal when a head attends to a token pair. In the body of the paper we show how we overcome these challenges to identify the signals that are causal for attention.

Once signals are identified, they can be used to address a number of challenges in model interpretability. First, they enable *precise* and *simple* circuit discovery (emphasized as an open problem in [8, 33]). We show a general method for using signals to identify the model components that are causal for model outputs. Across the models and tasks we study, the circuits we identify are all consistent with those found via previous methods. Each circuit obtained is specific to a given model input, and requires only a single model execution (without need for counterfactual inputs). Together, these aspects constitute an advance over the state of the art in circuit tracing.

Further, identifying signals uncovers and explains fundamental model mechanisms. Because the signals we identify are jointly determined by both model weights and runtime activations, they are not limited to data-dependent features. We broadly identify two classes: *data signals* and *control signals.* Control signals are present across many or all tokens, and are often used in the control of large groups of attention heads. This novel phenomenon occurs across models in our study.

**Related Work.**   Our work relates to mechanistic interpretability [34, 35], which seeks to understand neural network computations in human-understandable terms. Significant bodies of work in the area concern (a) how features are represented [22] and (b) how to trace circuits, ie, to identify model components that are responsible for solving specific tasks [23]. Our work contributes to both of these areas.

With respect to feature representation, much prior work has established the low-dimensionality of many feature representations [15, 22–32]. Feature representations have been extensively explored using Sparse Autoencoders (SAEs) [36, 37], but those methods have been shown to have significant drawbacks [38–42]; one reason for this can be that SAEs are built only from model activations, and do not take into account model parameters [43]. In contrast, our work jointly analyzes activations *and* model parameters (QK matrices) to identify features that are provably causal for attention. On the other hand, our work shares some ideas with [44], but that work is not based on causal analysis.

With respect to circuit tracing, much recent work has used methods based on counterfactual inputs ('activation patching') for establishing causality of model components [10, 11, 14, 45–47]. Patching is time-consuming, requiring many forward passes, generally requires the creation of a counterfactual data set to provide task-neutral activation patches, and exhibits a number of other weaknesses [48–51]. In this work, we trace circuits using only a single forward pass, eliminating the need for counterfactual inputs and avoiding the problems associated with patching.

Our method has similarities to but also important differences from [3]. Unlike that paper, we start with a formal problem definition of attention-causal communication and we present an algorithm that *provably* identifies attention-causal communication. Further, the method in [3] needs modification to work in general. This is because [3] assumed that moving the attention score of a token pair

towards zero will decrease the attention on the pair. However, this is not true in general (eg, when attention scores are negative), which is one motivation for our introduction of relative attention.

As noted above, our approach to signal identification builds on prior work showing that model components often communicate in low dimensional subspaces defined by the spectral decomposition of QK matrices [1–3]. Furthermore we show that the control signals we identify are proximal causes for attention sinks [52, 53] and generalize previous attention-sink mechanisms [54].

**Background.** In the model, token embeddings are $D$-dimensional, there are $H$ attention heads in each layer, and there are $L$ layers. We define $R = \frac{D}{H}$, which is the dimension of the spaces used for keys and queries in the attention mechanism. We use $N$ to denote the number of tokens in a given prompt. Superscript indices will denote (layer, head, destination token, source token); reduced sets of indices will be used where there is no confusion, and subscripts will generally denote matrix components.

To simplify exposition in the body of the paper, we consider models in which the attention mechanism does not apply a bias term to computation of keys and queries.[1] In Appendix B we describe how we handle models with bias terms in the QK circuit. Further, the body of the paper only discusses models with global positional encoding; in Appendix C we describe how we handle models that use rotary positional encoding [57] (RoPE). We emphasize that all the methods in the paper extend to models having attention bias terms and using RoPE, and we provide code [2] implementing our methods for those models.

The attention mechanism operates on a set of $N$ tokens in $D$-dimensional embeddings: $X \in \mathbb{R}^{N \times D}$. Each token $\mathbf{x} \in \mathbb{R}^D$ is passed through linear transforms given by $\mathbf{x}^\top W_K$, $\mathbf{x}^\top W_Q$, using weight matrices $W_K, W_Q \in \mathbb{R}^{D \times R}$. Then the inner product is taken for all pairs of transformed tokens to yield *attention scores*:

$$A'_{ds} = \mathbf{x}^{d\top} \Omega \mathbf{x}^s \tag{1}$$

in which $\Omega = W_Q W_K^\top$, $\mathbf{x}^d$ is the destination token, and $\mathbf{x}^s$ is the source token of the attention computation. We also refer to $\Omega$ as the head's QK matrix.

To enforce masked self-attention, $A'_{ds}$ is set to $-\infty$ for $d < s$. Attention scores are then normalized for each destination $d$, yielding *attention weights* $A_d = \text{Softmax}(A'_d / \sqrt{R})$. The resulting attention weight $A_{ds}$ is the amount of attention that destination $d$ is placing on source $s$. We denote the portion of the output of attention head $(\ell, a)$ that comes from $\mathbf{x}^s$ and is written into $\mathbf{x}^d$ via the OV-circuit as $\mathbf{o}^{\ell ads}$. Further details of model computations are given in Appendix A.

## 2 Signals

**Problem Statement.** As described in §1, our goal is to identify low-dimensional features that are causal for attention patterns. We define the problem of *attention-causal communication* as the following. For a given model and input, and given a head $(\ell, a)$ that attends to a token pair $(\mathbf{x}^d, \mathbf{x}^s)$, consider one of $\mathbf{x}^d$ or $\mathbf{x}^s$, denoted $\mathbf{x}$. Identify a $c$-dimensional subspace $C$, and a set $M$ consisting of $m$ model components upstream of $(\ell, a)$, with $c$ and $m$ as small as possible, such that if the components in $M$ had not written into $\mathbf{x}$ in subspace $\mathcal{C}$, head $(\ell, a)$ would not have attended to $(\mathbf{x}^d, \mathbf{x}^s)$.

To make our problem statement precise, we need to define when a head "attends to" a token pair. We adopt a very conservative definition: we say that a head $(\ell, a)$ attends to the token pair $(d, s)$ when $A^{\ell a}_{ds} \geq 1/n$, where $n$ is the number of tokens considered in the $\text{Softmax}$ calculation. That is, the head attends to the token pair when the attention it places on this pair is greater than what it would be for a uniform distribution over tokens. This could be considered too low a value in many cases – when the context window is large, the value of $1/n$ will be quite small. However, we emphasize that our methods can be used for any setting of attention threshold *greater* than $1/n$, and it is a simple matter to use a larger threshold in practice. The other phrase we need to make precise is for "$c$ and

---

[1]Models in this category include the Gemma [55] and Llama 3 [56] families.
[2]Code available at https://github.com/gaabrielfranco/pinpointing-attention-causal-communication

$m$ to be as small as possible." In practice we optimize these separately, first finding a subspace $\mathcal{C}$ having small dimension, and then finding the smallest set of upstream components ($M$) given $\mathcal{C}$.

We do not attempt to solve the attention-causal communication problem exactly, but instead present a heuristic method that gives very good results in practice. In the following sections we describe the method in detail.

**Relative Attention.** We first attack the problem of identifying the subspace $\mathcal{C}$. We isolate this problem as follows: consider a head $(\ell, a)$ and a token pair $(\mathbf{x}^d, \mathbf{x}^s)$, such that $A_{ds}^{\ell a} \geq 1/n$. Working for example with $\mathbf{x}^d$ (methods for $\mathbf{x}^s$ are analogous), we seek a low-dimensional subspace $\mathcal{C}$ such that in a counterfactual setting where the component of $\mathbf{x}^d$ in $\mathcal{C}$ were removed, we would have $A_{ds}^{\ell a} < 1/n$. That is, under the intervention $\mathrm{do}(\mathbf{x}^d = P_{\mathcal{C}\perp}\mathbf{x}^d)$, the head $(\ell, a)$ would not attend to the token pair $(\mathbf{x}^d, \mathbf{x}^s)$.

An additional constraint on $\mathcal{C}$ is that upstream components must write into $\mathbf{x}^d$ in $\mathcal{C}$. By virtue of the model's residual connections, we can consider $\mathbf{x}$ to be an initial encoding plus a sum of upstream additions [21]. Hence it is attractive to consider linear decompositions of $\mathbf{x}$ in determining $\mathcal{C}$. However, this strategy faces a key challenge: a head's attention $A_{ds}$ is a nonlinear function of $\Omega$, $\mathbf{x}^d$, and $\mathbf{x}^s$ through the $\mathrm{Softmax}$ function.

To address this problem, we introduce an ersatz function that usefully stands in for $\mathrm{Softmax}$ in causal analysis. Recall that $A_d = \mathrm{Softmax}(A_d'/\sqrt{R})$ as described in §1. We define the *relative attention* at head $(\ell, a)$ and position $(d, s)$ as:

$$c^{\ell a d s} = A_{ds}' - \frac{1}{d-1} \sum_{j \leq d,\, j \neq s} A_{dj}' \qquad (2)$$

where $A_{ds}' = \mathbf{x}^d \Omega^{\ell a} \mathbf{x}^s$. Relative attention has two important properties: first, it is linear in each of $\mathbf{x}^d$, $\Omega^{\ell a}$, and $\mathbf{x}^d$. The linearity of $c^{\ell a d s}$ in each of these will be important for our methods below.

The second important property of relative attention is that it is useful for causal analysis. In particular, we have the following Lemma:

**Lemma 1.** *Let $n \leq N$ be equal to the number of tokens considered in self-attention at head $(\ell, a)$ for destination token $d$, and $c^{\ell a d s}$ be defined as in (2). If $A_{ds}^{\ell a} \geq 1/n$, then $c^{\ell a d s} > 0$.*

We prove Lemma 1 in Appendix D. We can interpret Lemma 1 as saying that if an attention head puts more weight on a source than would occur under a uniform distribution, the relative attention (2) will be positive; likewise, when the relative attention is negative, the attention weight is less than what is given by the uniform distribution.

Using relative attention we can reframe the search for causal mediators of attention. Instead of looking for low-dimensional components of $\mathbf{x}$ that are causal for (nonlinear) attention greater than $1/n$, we can search for low-dimensional components that are causal for (linear) relative attention greater than zero.

**Attention Decomposition.** Relative attention is a useful tool in searching for attention-causal features, because it allows us to consider linear decompositions of $\mathbf{x}$ as candidate features. We first note that relative attention (2) can be expanded in the singular vectors of $\Omega$:

$$c^{\ell a d s} = \sum_{1 \leq k \leq R} \left( \mathbf{x}^{\ell d \top} \mathbf{u}^k \sigma^k \mathbf{v}^{k \top} \mathbf{x}^{\ell s} - \frac{1}{d-1} \sum_{j \leq d,\, j \neq s} \mathbf{x}^{\ell d \top} \mathbf{u}^k \sigma^k \mathbf{v}^{k \top} \mathbf{x}^{\ell j} \right) \qquad (3)$$

where $\sum_{1 \leq k \leq R} \mathbf{u}^k \sigma^k \mathbf{v}^{k \top}$ is the singular value decomposition of $\Omega^{\ell a}$. Then the following hypothesis (adapted from [3]) drives our approach:

**Hypothesis (Sparse Attention Decomposition [3])** When an attention head performs a task that requires detecting components in a pair of low-dimensional subspaces in its inputs $\mathbf{x}^d$ and $\mathbf{x}^s$, and its inputs have significant components in those subspaces, the terms in (3) will show large values for a distinct subset of values of $k$.

In §3 we demonstrate that sparse attention decomposition occurs widely across the models, tasks, and attention heads we examine. Typically the number of significant terms in (3) is 20 or less, and

is often as small as 5. This gives us leverage on determining $\mathcal{C}$. When considering an attention head $(\ell, a)$ that is attending to tokens $(\mathbf{x}^d, \mathbf{x}^s)$, we will have that $c^{\ell ads}$ will be positive. We then can identify $\mathcal{C}$ as defined by the smallest set of terms responsible for the positive value of $c^{\ell ads}$, and by the sparse attention decomposition hypothesis, we expect that $\mathcal{C}$ will typically be a low-dimensional subspace. This strategy builds on prior work [1–3] showing that the singular vectors of QK matrices provide useful decompositions of attention and inter-layer communication.

**Why Should Attention Show Sparse Decomposition?**  Here we pause to ask: why should attention show sparse decomposition? To simplify matters, we consider just the bilinear form $\mathbf{x}^{d\top} \Omega \mathbf{x}^s$ and ask why it should show sparse decomposition in its singular vectors.

We start by noting that for a square matrix $A$, a consequence of the Courant-Fischer theorem is that, over all unit vectors $\mathbf{x}$, the maximum value of the quadratic form $\mathbf{x}^\top A \mathbf{x}$ is obtained when $\mathbf{x}$ is the principal eigenvector of $A$. This is readily extended to bilinear forms: over all unit vectors $\mathbf{x}$ and $\mathbf{y}$, the maximum value of $\mathbf{x}^\top A \mathbf{y}$ is obtained when $\mathbf{x}$ is the principal left singular vector, and $\mathbf{y}$ the principal right singular vector, of $A$. From the standpoint of model training, another view is useful:

**Lemma 2.** *Given vectors $\mathbf{x}$ and $\mathbf{y}$, among all rank-1 matrices having unit Frobenius norm, the matrix $D$ that maximizes $\mathbf{x}^\top D \mathbf{y}$ is $D = \frac{\mathbf{x}}{\|\mathbf{x}\|} \frac{\mathbf{y}^\top}{\|\mathbf{y}\|}$.* (Proof is provided in Appendix E.)

In a model training setting, we can hypothesize (following [15, 22–32]) that $\mathbf{x}^d$ and $\mathbf{x}^s$ contain corresponding feature sets $\{\phi^i\}$, $\{\gamma^i\}$ that are encoded in one-dimensional or in low-dimensional subspaces. For example, feature $\phi^0 \in \mathbb{R}^D$ could correspond to a geographic location encoded in $\mathbf{x}^d$, as in [27]. We further note that the authors in [22] argue that models will tend to represent correlated feature sets in a manner such that, considered in isolation, the sets are *nearly* orthogonal. They term this the use of "local, almost-orthogonal bases." In our case, if an attention head with QK matrix $\Omega$ is tuned to detect feature sets $\{\phi^i\}$ and $\{\gamma^i\}$ that are important when performing a specific task, then we may hypothesize that training will construct the sets to be "nearly-orthogonal," meaning that cosine similarities among the features in each set would typically be small. In this case, Lemma 2 suggests that the learned $\Omega$ will have sets of singular vectors $\{\mathbf{u}_k\}$, $\{\mathbf{v}_k\}$ that are likely to sparsely encode the $\{\phi^i\}$ and $\{\gamma^i\}$.

**Isolating Signals.**  We can now describe how to decompose residuals to separate causal signals they contain from background. The goal is to identify which terms in (3) form the sparse representation of $c^{\ell ads}$, which is akin to denoising in signal processing. Starting from a head $(\ell, a)$ attending to token pair $(\mathbf{x}^d, \mathbf{x}^s)$, we have that $A_{ds}^{\ell a} \geq 1/n$, and so by Lemma 1 $c^{\ell ads} > 0$. We separate the terms in (3) into two sets, one of which is the smallest set whose sum exceeds $c^{\ell ads}$, and the other whose sum is close to (but less than) zero.

Specifically, we define $\mathcal{S}^{\ell ads}$ as (the indices of) the smallest set of terms in (3) that in sum exceed $c^{\ell ads}$. The terms captured in $\mathcal{S}^{\ell ads}$ are strictly positive, and are the largest positive terms in (3). Given $\mathcal{S}^{\ell ads}$, we can decompose model residuals into 'signal' and 'noise' in terms of their impact on attention scores as measured by $c^{\ell ads}$. Define subspaces $\mathcal{U} = \text{Span}\{\mathbf{u}^k \,|\, k \in \mathcal{S}^{\ell ads}\}$ and $\mathcal{V} = \text{Span}\{\mathbf{v}^k \,|\, k \in \mathcal{S}^{\ell ads}\}$ and associated projectors $P_{\mathcal{U}}$ and $P_{\mathcal{V}}$. The denoising step separates the residuals $\mathbf{x}^{\ell d}$ and $X^\ell$ into:

$$\mathbf{s}^{\ell d} = P_{\mathcal{U}} \mathbf{x}^{\ell d}, \quad \mathbf{z}^{\ell d} = P_{\mathcal{U}^\perp} \mathbf{x}^{\ell d}, \quad S^\ell = X^\ell P_{\mathcal{V}}^\top, \quad Z^\ell = X^\ell P_{\mathcal{V}^\perp}^\top, \tag{4}$$

where $P_{\mathcal{U}^\perp} = I - P_{\mathcal{U}}$ and $P_{\mathcal{V}^\perp} = I - P_{\mathcal{V}}$. Then we have $\mathbf{x}^{\ell d} = \mathbf{s}^{\ell d} + \mathbf{z}^{\ell d}$, $X^\ell = S^\ell + Z^\ell$, and

$$\mathbf{s}^{\ell d\top} \Omega^{\ell a} S_s^\ell - \frac{1}{d-1} \sum_{j \leq d, j \neq s} \mathbf{s}^{\ell d\top} \Omega^{\ell a} S_j^\ell \approx c^{\ell ads} \quad \text{and} \quad \mathbf{z}^{\ell d\top} \Omega^{\ell a} Z_s^\ell - \frac{1}{d-1} \sum_{j \leq d, j \neq s} \mathbf{z}^{\ell d\top} \Omega^{\ell a} Z_j^\ell \leq 0, \tag{5}$$

where $|\mathcal{S}^{\ell ads}| = \dim \mathcal{U} = \dim \mathcal{V}$ is as small as possible. Note that by definition, under the counterfactual in which signal $\mathbf{s}^{\ell d}$ is not present in $\mathbf{x}^{\ell d}$ (i.e., $\text{do}(\mathbf{x}^{\ell d} = P_{\mathcal{U}^\perp} \mathbf{x}^{\ell d})$) then $c^{\ell ads} < 0$, and so $(\ell, a)$ would *not* have attended to token pair $(\mathbf{x}^d, \mathbf{x}^s)$. The corresponding conclusion holds as well for $S^\ell$ and $X^\ell$.

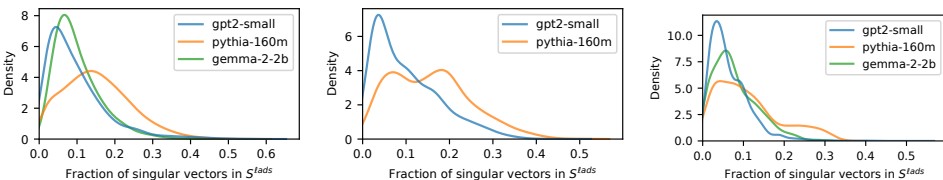

Figure 1: Dimension of signal subspace compared to $R$ in (left to right): IOI task, GT task, GP task.

# 3 Sparse Attention Decomposition

Next we confirm prior work showing that the phenomenon of sparse attention decomposition is ubiquitous [1–3]. We utilize models spanning various architectures and scales: GPT-2 small [19], Pythia-160M [58], and Gemma-2 2B [55]. These models are evaluated on multiple tasks: Indirect Object Identification (IOI) [45], the Greater Than (GT) task[3] [59], and Gender Pronoun (GP) [60]. These models and tasks cover a wide range of studies in mechanistic interpretability, specifically in circuit discovery [14, 61–63].

We first determine $\mathcal{S}^{\ell ads}$ for significant $A_{ds}^{\ell a} > \frac{1}{n}$ values using the strategy in §2. We then count the number of singular vectors utilized from Equation (3) for each relevant head and token pair. Figure 1 illustrates the resulting distributions of the fraction of singular vectors used to construct $\mathcal{S}^{\ell ads}$, ie, the distribution of $|\mathcal{S}^{\ell ads}|/R$. The figure illustrates the sparsity of attention decomposition across different attention heads, prompts, models, and tasks. Typically, models use only 5–10% of the available $R$ dimensions (e.g., 3–6 for GPT-2/Pythia, 13–25 for Gemma-2)—and almost always less than half—to compute their attention scores. Additional plots demonstrating this sparse attention decomposition for individual attention heads in various tasks are in Appendix G. These findings confirm that sparse attention decomposition is widespread, indicating that signals generally inhabit low-dimensional subspaces.

# 4 Application: Tracing Communication and Circuits

The results of §3 suggest an application to tracing communication. In this section we describe how one may use signals to trace communication within the model.

**Causal Structure.** To start, we note that the use of residual connections in the model makes it possible to describe a residual $\mathbf{x}^d$ at the input to layer $\ell$ in the model as the following sum:

$$\mathbf{x}^{\ell d} \in \mathbb{R}^D = \overbrace{\sum_{1 \le l < \ell} \sum_{1 \le a \le H} \sum_{1 \le s \le N} \mathbf{o}^{lads}}^{\text{AH outputs}} + \overbrace{\sum_{1 \le l < \ell} \mathbf{f}^{ld}}^{\text{FFN outputs}} + \overbrace{\sum_{1 \le l < \ell} \mathbf{b}_O^l}^{\text{attn. biases}} + \overbrace{\mathbf{x}^{0d}}^{\text{input at layer 0}} \tag{6}$$

where $\mathbf{o}^{lads}$ is defined in §1, $\mathbf{f}^{ld}$ is the output of the FFN at layer $l$ for resdual $d$, $\mathbf{b}_O^l$ is the bias term for layer $l$, and $\mathbf{x}^{0d}$ is the input embedding of token $d$. Equation (6) provides a complete and exact decomposition of a residual at any layer in the model.

To characterize the causal structure of the model with respect to relative attention, we define two functions $d(\cdot)$ and $s(\cdot)$ parameterized by a head $(\ell, a)$ and a token pair $(\mathbf{x}^d, \mathbf{x}^s)$. These functions measure the extent to which a particular model component affects the relative attention $c^{\ell ads}$. The function $d^{\ell ads}(\mathbf{w})$ measures what the relative attention $c^{\ell ads}$ would be if the source tokens were fixed as $X$ while $\mathbf{w}$ had been the destination token; and $s^{\ell ads}(W)$ if the set of source tokens had been $W$ but the destination token were fixed as $\mathbf{x}^d$ (formal definitions are in Appendix H). Each function is linear in its argument. Note that

$$c^{\ell ads} = d^{\ell ads}(\mathbf{x}^{\ell d}) = s^{\ell ads}(X^\ell). \tag{7}$$

To characterize causality in the model with respect to relative attention, we distribute (7) over (6) as described in Appendix H. This gives, for each upstream component, its contribution to the relative attention at head $(\ell, a)$ and token pair $(\mathbf{x}^d, \mathbf{x}^s)$.

---

[3]Gemma-2 2B was not analyzed for the GT task, as its distinct number tokenization (requiring two sequential tokens for two-digit numbers) complicates performance comparisons; details are in the Appendix F.

**Tracing Attention-Causal Communication.** The pieces are now in place to trace causal communication within the model. Our goal is, given $A_{ds}^{\ell a} \geq 1/n$, find the upstream components that are responsible for head $(\ell, a)$ attending to token pair $(\mathbf{x}^d, \mathbf{x}^s)$.

We describe the tracing process starting from a destination token $\mathbf{x}^{\ell d}$. The process starting from a source token is analogous, and we provide full details in Appendix H.

Using the methods of §2, we obtain the subspace $\mathcal{U}$ in which the signals reside. This allows us to isolate signals as in (4), obtaining $\mathbf{s}^{\ell d}$. By (5), (7) and the properties of the SVD, we have

$$c^{\ell ads} \approx d^{\ell ads}(\mathbf{s}^{\ell d}) = d^{\ell ads}(P_{\mathcal{U}}\mathbf{x}^{\ell d}). \tag{8}$$

To identify the upstream components writing these signals into the residuals, we distribute (8) over (6). Then we can determine the upstream contributions to $c^{\ell ads}$ by making use of the linearity of $d(\cdot)$ and the projection operations. Specifically, distributing (8) over (6) results in the upstream contributions:

$$d_{lhdt}^{\ell ads} = d^{\ell ads}(P_{\mathcal{U}}\mathbf{o}^{lhdt}), \quad d_{ld}^{\ell ads} = d^{\ell ads}(P_{\mathcal{U}}\mathbf{f}^{ld}), \quad d_l^{\ell ads} = d^{\ell ads}(P_{\mathcal{U}}\mathbf{b}_O^l), \quad d_0^{\ell ads} = d^{\ell ads}(P_{\mathcal{U}}\mathbf{x}^{0d}),$$

(where subscripts here denote upstream indices), allowing us to decompose the relative attention in terms of each upstream component's contribution:

$$c^{\ell ads} \approx \sum_{l<\ell, h \leq H, t \leq N} d_{lhdt}^{\ell ads} + \sum_{l<\ell} d_{ld}^{\ell ads} + \sum_{l<\ell} d_l^{\ell ads} + d_0^{\ell ads} \tag{9}$$

We provide more details, and an equivalent decomposition for the source token, in Appendix H. Equation (9) shows how to measure the contribution of each upstream component to the downstream relative attention. Each term in Equation (9) represents an edge in the communication graph of the model's computation.

To build a communication graph for a given model and prompt, we start by finding the heads that contribute the largest signal in the direction of the model output. Then, for any of those heads that are attending to a token pair and writing in that direction, we find the upstream components (attention heads, MLPs, bias terms, or input tokens) causal for their attention. We then work backward, recursively tracing signals from downstream to upstream. The recursion terminates on any head in the first layer, or that is not attending to a token pair, or on an MLP, bias term, or input token. Our algorithm is described in detail in Appendix H, and the code for all our methods is available.

**Communication Graphs.** We construct communication graphs for GPT-2 small, Pythia-160M, and Gemma-2 2B across the IOI, GP, and GT tasks. Appendix H provides details of the process, running times, and examples of resulting communication graphs. All experiments were conducted on CPU-only machines.

Communication graphs (Appendix H) are highly detailed and contain extensive information about information flow in a model's computation. To illustrate, we present aggregated versions for the IOI task in Figure 3, in which each edge's thickness reflects the number of edges between tokens in the full communication graph. Even aggregated, these graphs illustrate differences in the task solution strategies used by the three models. For example, they confirm that GPT and Pythia rely on identifying duplication of the subject token ('Simon'), and later tokens receive information from both occurrences (as reported in [45, 63]). In contrast, Gemma uses the token 'Andrew' as an information-aggregating anchor, with subsequent tokens receiving information from this anchor. Complete communication graphs for these prompts are in the Appendix.

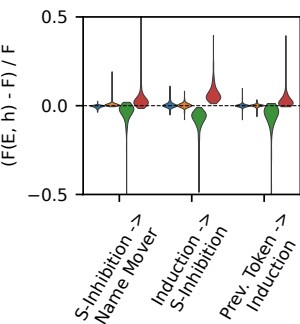

Figure 2: Intervention effect, GPT-2/IOI. Green: signal ablation; Red: signal boosting; Blue: random ablation; Orange: random boosting.

**Interventions.** To validate our communication graphs, we demonstrate that intervening in the signals we identify has a causal impact on model performance. For any signal $\mathbf{s}$, we boost it $(\text{do}(\mathbf{x} = \mathbf{x} + \mathbf{s}))$ or ablate it $(\text{do}(\mathbf{x} = \mathbf{x} - \mathbf{s}))$ for the relevant token $\mathbf{x}$ at input to the relevant attention head $(\ell, a)$ [4]. We then measure the performance of the model on the given task. Performance is task-specific, but is measured in all cases via relative logit differences between correct and incorrect outputs, with positive values indicating improved model performance.

---

[4]Specifically, we intervene only on the input of either the Q or K transformation.

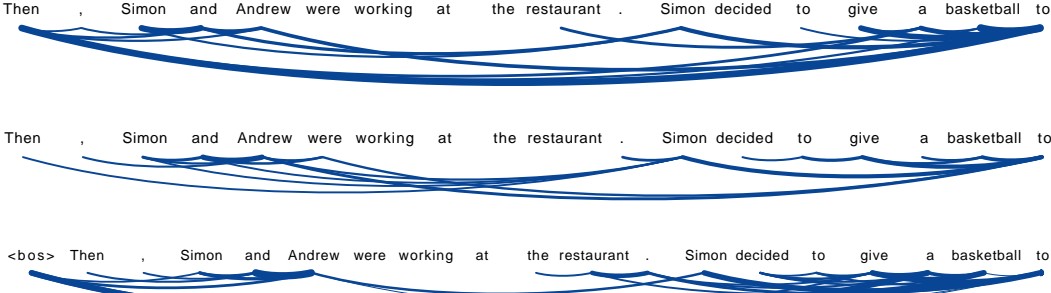

Figure 3: Aggregated communication graphs, IOI task. Top to bottom: GPT-2, Pythia, Gemma-2.

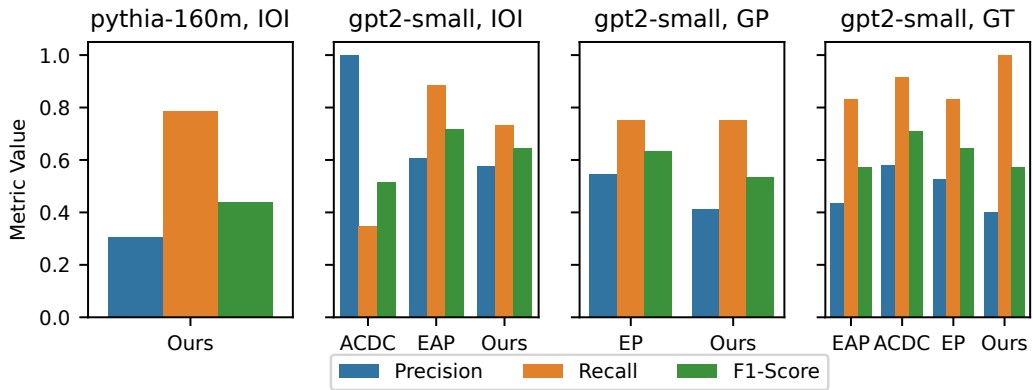

Figure 4: Circuit performance. First-reported circuit used for comparison for every task/model.

Figure 2 shows typical results; full results are in Appendix J. We find that ablating any of the test signals leads to performance decreases, and boosting the signal leads to performance improvements, across the three models and three tasks. Note also that these interventions have an extremely small impact on residuals themselves, with an average relative change in vector norm less than 1% and cosine similarity before and after typically greater than 0.99 (see Appendix J).

**Circuits.** To find circuits (maximal collections of model components causal for performance), we aggregate communication graphs over multiple prompts, threshold edges based on their causal impact on model output, and remove components not causal for the output. For comparative analysis, we focus on methods that report full circuit components, such as Edge Pruning (EP) [61], ACDC [14], and EAP [62]. We note that reported circuits for each task vary considerably and lack definitive ground-truth, so we seek to know whether our circuits are broadly consistent with those reported by others. To choose a baseline for comparison, we use the first-reported circuit for each task and model combination (Pythia/IOI: [63], GPT-2/IOI: [45], GPT-2/GP: [60], and GPT-2/GT: [59]), against which we compute precision, recall, and $F_1$-scores for all other methods. Figure 4 shows that our approach reports circuits that are consistent with previously reported circuits. Details and more comparisons are in Appendix K.

## 5 Application: Control Signals

As a second application of our methods, we examine all signals used in a single forward pass (no longer limiting to those in the communication graph as in §4). We find that many signals are *data-independent* – they are used by heads in many layers, and they are present in many or all tokens. These signals play a role in organizing model computation; we call them *control signals*.

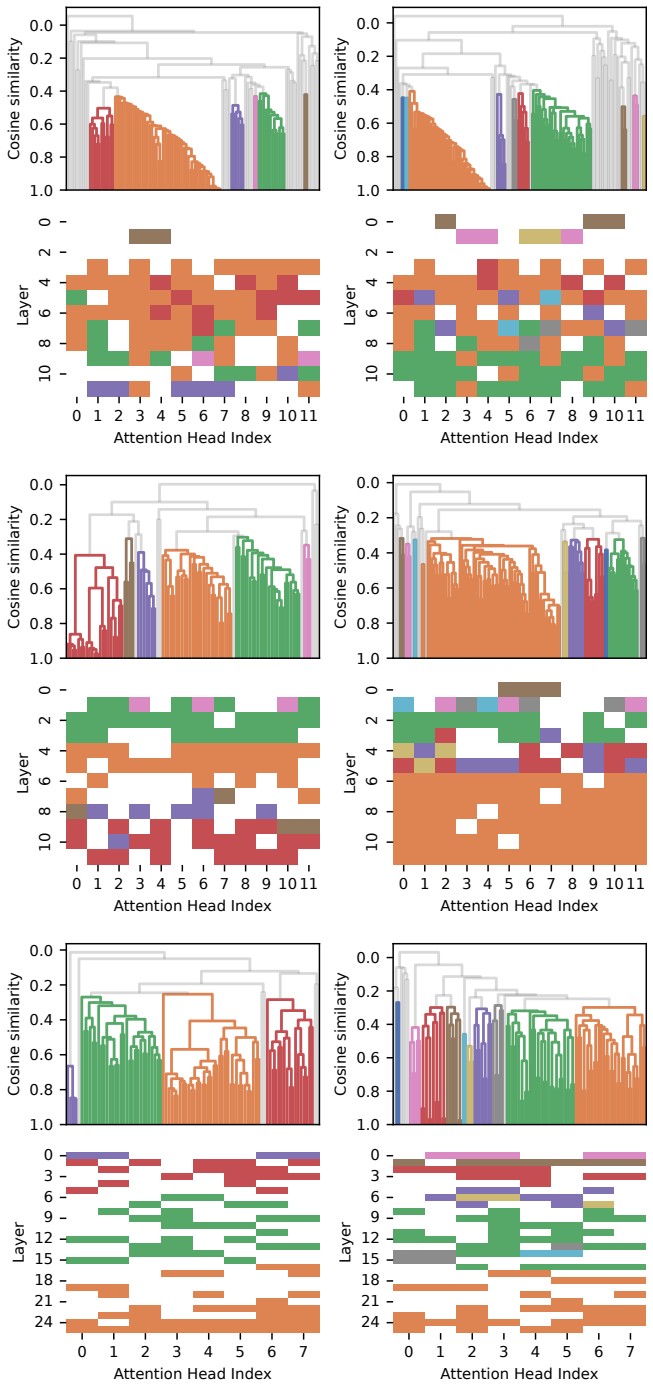

Figure 5: Control signals and associated heads: top: GPT-2, middle: Pythia, bottom: Gemma-2.

Control signals are easily identified; in each of the models we study, two clear clusters of signals are present, with the larger cluster corresponding to control signals. The functional distinction between control signals and data signals concerns the source token that the head is attending to. When the source token is the first token (in GPT-2), the first token or a punctuation token (in Pythia and Gemma), the signal that is causal for attention is from the control cluster. When the source token is not one of those, the causal signal is a data signal.

Hence, control signals are the mechanism implementing *attention sink* behavior [52–54] at the signal level. The attention sink phenomenon is the tendency of models to attend to an initial token when the head does not need to move information into the destination token. The Softmax operation imposes an attention distribution over sources, so the head uses a semantically unimportant token as source in these situations. We observe that typically more than 80% of all token pairs attended to in our experiments are due to attention sink.

Control signals come in pairs: one that resides in the relevant source (eg, start) token, and one that resides in every token as a potential destination. We find that control signals are the causal mechanism for attention sinking; they are strongly predictive of attention-sink behavior, with F-scores of 0.965 (GPT-2), 0.952 (Pythia), and 0.994 (Gemma-2). In intervention experiments akin to those in §4, we observe that boosting the control signal in either source or target tokens causes the head to revert to attention sinking, essentially "shutting it down," regardless of the tokens presented.

Next, we show that most control signals belong to one of a small number of clusters and that attention heads are functionally organized according to those clusters. In Figure 5 the top row shows signal clusters (source left, destination right) for the three models. Height corresponds to cosine similarity; note that a similarity greater than 0.2 is quite significant, and most clusters have internal similarity greater than 0.4. Hence, there are only a few really distinct control signals used in these models – less than a dozen. More surprisingly, these distinct control signals generally are used in different parts of the model. The second row shows where each control signal from the top row is used in the model. We see that models are hierarchically organized into groups of heads that tend to share a common control signal. The functional significance of this organization is an intriguing direction for further study.

Finally, we ask how models manage the control signals in tokens. We observe that control signals are added in layer 0, either by the attention head layer or the MLP. Over the layers of the model, signal strength tends to rise towards the layers where the signal is used (as shown in lower row in Figure 5). As expected, source control signals have a strongly negative cosine similarity with non-start tokens, and vice versa; this is to avoid confusion in the attention sink process. Figures 24 and 25 in Appendix M show how control signals vary over the layers of the models.

## 6  Conclusions

In this paper we make a number of contributions. We define the attention-causal communication problem, and we develop a theoretically grounded, heuristic approach to address it. Our approach is based on two key ideas: first, singular vectors of the QK matrix should in some sense 'match' relevant features in residuals; and second, relative attention (which we define) provides a causally-useful linear ersatz for the nonlinear attention computation. We build on previous observations that attention is generally sparsely decomposable in the QK basis, which leads to an efficient method for identifying low-dimensional subspaces in which signals reside. These signals enable new, precise methods to expose important communication with the model, and expose mechanisms of global model coordination.

**Limitations.** Our approach, while promising, shares the quadratic complexity in the number of tokens inherent in standard attention mechanisms. Additionally, the proposed solution to the attention-causal communication problem is a heuristic. While empirically effective, future work could explore more formally guaranteed methods as well as causality involving MLPs.

## Acknowledgments and Disclosure of Funding

This research was funded by a grant from Open Philanthropy and by NSF award CNS-2312711. Code for our method is available at https://github.com/gaabrielfranco/pinpointing-attention-causal-communication; its implementation was enabled by the TransformerLens library [64]. We thank Evimaria Terzi, Aaron Mueller, and members of our research group at Boston University for helpful discussions, and we thank the anonymous NeurIPS referees for their feedback which improved this paper.

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

# A  Further Background

Here we provide further background on model computations and associated notation.

To compute the output of the attention head each input token is first passed through an affine transformation: $V = XW_V + \mathbf{1}\mathbf{b}_V^\top$ with $W_V \in \mathbb{R}^{D \times R}, \mathbf{b}_V \in \mathbb{R}^R$. Attention weights are then used to combine rows of $V$ to construct the attention head's outputs: $Z = AV$, with $Z \in \mathbb{R}^{N \times R}$. Next each head's output is passed through a linear transformation yielding the per-head output $O = ZW_O$ which transforms it back into the $D$-dimensional embedding space, with $O \in \mathbb{R}^{N \times D}$. For a given layer $\ell$ the final output of the attention block is then the sum over heads plus a per-layer bias:

$$O^\ell = \sum_{1 \le a \le H} O^{\ell a} + \mathbf{1}\mathbf{b}_O^\top \tag{10}$$

For a given attention head, we will at times need to decompose the per-head output $O^{\ell a} = AVW_O$ into the portions contributed by each source token $s$. Let $\mathbf{o}^{\ell a d} \in \mathbb{R}^D$ denote the output of head $(\ell, a)$ for token $d$. This corresponds to row $d$ of $O^{\ell a}$. Let $A_d \in \mathbb{R}^N$ be the $d$-th row of the $(\ell, a)$ attention matrix, $V \in \mathbb{R}^{N \times R}$ be the value computation in head $(\ell, a)$, and $W_O \in \mathbb{R}^{R \times D}$ be the output matrix of head $(\ell, a)$. Then the per-source output is:

$$\mathbf{o}^{\ell a d s} = A_{ds}V_sW_O \tag{11}$$

where $V_s$ is row $s$ of $V$. Note that we have $\mathbf{o}^{\ell a d} = \sum_{1 \le s \le N} \mathbf{o}^{\ell a d s}$, with $\mathbf{o}^{\ell a d s} = \mathbf{0}$ when $s > d$ due to masked self-attention.

**SVD.**  Our methods make use of the SVD of $\Omega$. The matrix $\Omega$ has size $D \times D$, but due to its construction it has maximum rank $R$. We therefore work with the SVD of $\Omega = U\Sigma V^\top$ in which $U \in \mathbb{R}^{D \times R}$, $V \in \mathbb{R}^{D \times R}$ and $\Sigma \in \mathbb{R}^{R \times R}$. $U$ and $V$ are orthonormal matrices with $U^\top U = I$ and $V^\top V = I$, and $\Sigma = \text{diag}(\sigma_1, \sigma_2, \ldots, \sigma_R)$ with $\sigma_1 \ge \sigma_2 \ge \cdots \ge \sigma_R \ge 0$. Important to our work is that the SVD of $\Omega$ can equivalently be written as

$$\Omega = \sum_{k=1}^{R} \mathbf{u}^k \sigma^k \mathbf{v}^{k\top} \tag{12}$$

in which $\{\mathbf{u}^k\}$ and $\{\mathbf{v}^k\}$ are orthonormal sets and each term in the sum is a rank-1 matrix having Frobenius norm $\sigma_k$.

# B  Bias in Attention

Here we discuss how to handle models in which the attention head's key-query computation includes bias terms.

In such a model, each token $\mathbf{x} \in \mathbb{R}^D$ is passed through two affine transforms given by $\mathbf{x}^\top W_K + \mathbf{b}_K^\top$, $\mathbf{x}^\top W_Q + \mathbf{b}_Q^\top$, using weight matrices $W_K, W_Q \in \mathbb{R}^{D \times R}$ and offsets $\mathbf{b}_K, \mathbf{b}_Q \in \mathbb{R}^R$. Then the inner product is taken for all pairs of transformed tokens to yield attention scores. Specifically:

$$
\begin{aligned}
A'_{ds} &= (\mathbf{x}^{d\top}W_Q + \mathbf{b}_Q^\top)(\mathbf{x}^{s\top}W_K + \mathbf{b}_K^\top)^\top \\
&= \mathbf{x}^{d\top}W_QW_K^\top\mathbf{x}^s + \mathbf{x}^d W_Q\mathbf{b}_K + \mathbf{b}_Q^\top W_K^\top\mathbf{x}^s + \mathbf{b}_Q^\top\mathbf{b}_K
\end{aligned}
\tag{13}
$$

in which $\mathbf{x}^d$ is the destination token and $\mathbf{x}^s$ is the source token of the attention computation.

As a result, for models with bias we will use two versions of (13). We define $\Omega_d = [W_QW_K^\top \ W_Q\mathbf{b}_K]$, $\Omega_s = \begin{bmatrix} W_QW_K^\top \\ \mathbf{b}_Q^\top W_K \end{bmatrix}$, and we use $\tilde{\mathbf{x}}$ to denote $\begin{bmatrix} \mathbf{x} \\ 1 \end{bmatrix}$. Then:

$$A'_{ds} = \mathbf{x}^{d\top}\Omega_d\tilde{\mathbf{x}}^s + \mathbf{b}_Q^\top W_K\mathbf{x}^s + \mathbf{b}_Q^\top\mathbf{b}_K, \tag{14a}$$

$$A'_{ds} = \tilde{\mathbf{x}}^{d\top}\Omega_s\mathbf{x}^s + \mathbf{x}^{d\top}W_Q\mathbf{b}_K + \mathbf{b}_Q^\top\mathbf{b}_K. \tag{14b}$$

Given these new definitions, when expanding (2) using (14b) we still have that:

$$c^{\ell ads} = \tilde{\mathbf{x}}^{\ell d\top}\Omega_s^{\ell a}\mathbf{x}^{\ell s} - \frac{1}{d-1}\sum_{\substack{j\le d \\ j\ne s}} \tilde{\mathbf{x}}^{\ell d\top}\Omega_s^{\ell a}\mathbf{x}^{\ell j}. \tag{15}$$

However, when expanding (2) using (14a) we have that

$$c^{\ell ads} = \mathbf{x}^{\ell d\top}\Omega_d^{\ell a}\tilde{\mathbf{x}}^{\ell s} - \frac{1}{d-1}\sum_{\substack{j\le d \\ j\ne s}} \mathbf{x}^{\ell d\top}\Omega_d^{\ell a}\tilde{\mathbf{x}}^{\ell j} + c_1 \tag{16}$$

where $c_1 = \mathbf{b}_Q^\top W_K\mathbf{x}^{\ell s} - \frac{1}{(d-1)}\sum_{j\ne s}\mathbf{b}_Q^\top W_K\mathbf{x}^{\ell j}$. To understand the significance of this term, note that $c_1$ is only a function of source tokens; it arises due to the inner products of source keys with the constant query bias.

When determining the subspaces corresponding to signals as described in §2, we use the SVDs of $\Omega_d$ or $\Omega_s$ as appropriate to the setting. Specifically, when determining $P_{\mathcal{U}}$ we use

$$c^{\ell ads} - c_1 = \sum_{1\le k\le R}\left(\mathbf{x}^{\ell d\top}\mathbf{u}_k\sigma_k\mathbf{v}_k^\top\tilde{\mathbf{x}}^{\ell s} - \frac{1}{d-1}\sum_{\substack{j\le d \\ j\ne s}}\mathbf{x}^{\ell d\top}\mathbf{u}_k\sigma_k\mathbf{v}_k^\top\tilde{\mathbf{x}}^{\ell j}\right) \tag{17}$$

where the SVD of $\Omega_d$ is used. Note that we can ignore the $c_1$ term because it only arises due to source tokens, and so adds a constant that can be ignored when defining $P_{\mathcal{U}}$ for decomposing the destination token. The terms from (17) are chosen using same rule as in §2, namely the smallest set that equals or exceeds $c^{\ell ads} - c_1$; these terms are used to define $P_{\mathcal{U}} \in \mathbb{R}^{D\times D}$ in terms of the left singular vectors $\{\mathbf{u}_k\}$ of $\Omega_d$. When determining $P_{\mathcal{V}}$ we use

$$c^{\ell ads} = \sum_{1\le k\le R}\left(\tilde{\mathbf{x}}^{\ell d\top}\mathbf{u}_k\sigma_k\mathbf{v}_k^\top\mathbf{x}^{\ell s} - \frac{1}{d-1}\sum_{\substack{j\le d \\ j\ne s}}\tilde{\mathbf{x}}^{\ell d\top}\mathbf{u}_k\sigma_k\mathbf{v}_k^\top\mathbf{x}^{\ell j}\right) \tag{18}$$

where the SVD of $\Omega_s$ is used. Here again we select the smallest set of terms that equals or exceeds $c^{\ell ads}$ and use those to determine $P_{\mathcal{V}}$ in terms of the right singular vectors of $\Omega_s$.

To perform communication tracing as described in §4 requires the following adjustment to (7):

$$d^{\ell ads}(\mathbf{w}) = \mathbf{w}^\top\Omega_d^{\ell a}\tilde{\mathbf{x}}^{\ell s} - \frac{1}{d-1}\sum_{\substack{j\le d \\ j\ne s}}\mathbf{w}^\top\Omega_d^{\ell a}\tilde{\mathbf{x}}^{\ell j} \tag{19}$$

and

$$s^{\ell ads}(W) = \tilde{\mathbf{x}}^{\ell d\top}\Omega_s^{\ell a}W_s - \frac{1}{d-1}\sum_{\substack{j\le d \\ j\ne s}}\tilde{\mathbf{x}}^{\ell d\top}\Omega_s^{\ell a}W_j \tag{20}$$

With these new definitions, we still have

$$c^{\ell ads} = s^{\ell ads}(X^\ell) \tag{21}$$

however, we also have

$$c^{\ell ads} = d^{\ell ads}(\mathbf{x}^{\ell d}) + c_1. \tag{22}$$

Hence to perform communication tracing on the destination token, we look for the smallest set of upstream components whose portions of the contribution sum to $c^{\ell ads} - c_1$. Here again we ignore the $c_1$ term because it arises due to source tokens, and we are tracing inputs to the destination token.

Finally, one complication that arises is that it can happen that even though $A_{ds}^{\ell a} \ge 1/n$ for some token pair $(d, s)$, the quantity $d^{\ell ads}(\mathbf{x}^{\ell d}) = c^{\ell ads} - c_1$ is not positive. If this occurs, it is because the bias term $c_1$ is responsible for the positive value of $c^{\ell ads}$ leading to $A_{ds}^{\ell a} \ge 1/n$. In this case, we simply ignore the fact that $A_{ds}^{\ell a} \ge 1/n$ – ie, we do not compute signals, nor trace upstream from this $(\ell, a, d, s)$ tuple.

## C    RoPE

In models using Rotary Position Encoding (RoPE) [57], before computing the attention score, each token has a set of rotations performed that depend on the token's position in the prompt. The result is that for a given pair of tokens $(d, s)$, we can capture the effect of RoPE using a rotation matrix $R^{(d-s)} \in \mathbb{R}^{R \times R}$, where $(d - s)$ is used to denote the relative positions of the $d$ and $s$ tokens in the prompt. As a result, we must define $\Omega$ with respect to $(d - s)$. Specifically, we define

$$\Omega^{(d-s)} = W_Q R^{(d-s)} W_K.$$

Then we can rewrite the attention score computation as

$$A'_{ds} = \mathbf{x}^{d\top} \Omega^{(d-s)} \mathbf{x}^s \tag{23}$$

(All the work in this section we will be in the context of given head $(\ell, a)$ and we will drop the indices for the head – so for example $c^{\ell ads}$ will be written as just $c^{ds}$.)

Expanding (2) in terms of (23) we have that:

$$c^{ds} = \mathbf{x}^{d\top} \Omega^{(d-s)} \mathbf{x}^s - \frac{1}{d-1} \sum_{\substack{j \leq d \\ j \neq s}} \mathbf{x}^{d\top} \Omega^{(d-j)} \mathbf{x}^j \tag{24}$$

in which we have dropped a constant term that is handled similarly to $c_1$ in Appendix B.

It's no longer possible to write (3) in its given form because $c^{ds}$ involves multiple, different $\Omega$ matrices. However, our goal is still to decompose the contribution $c^{ds}$ in terms of the singular vectors of $\Omega^{(d-s)}$. Hence we proceed by defining the projection matrices associated with each singular vector of $\Omega^{(d-s)}$: $P_{u_k} = \mathbf{u}_k^{(d-s)} (\mathbf{u}_k^{(d-s)})^\top$ and $P_{v_k} = \mathbf{v}_k^{(d-s)} (\mathbf{v}_k^{(d-s)})^\top$. Because $\{\mathbf{u}_k^{(d-s)}\}$ and $\{\mathbf{v}_k^{(d-s)}\}$ are orthonormal bases for $\mathbb{R}^D$, we have that $\mathbf{x} = \sum_k P_{v_k} \mathbf{x} = \sum_k P_{u_k} \mathbf{x}$ for any $\mathbf{x}$. We can now formulate (3) in two equivalent ways:

$$c^{ds} = \left( \sum_{1 \leq k \leq R} P_{u_k} \mathbf{x}^d \right)^\top \Omega^{(d-s)} \mathbf{x}^s - \frac{1}{d-1} \sum_{\substack{j \leq d \\ j \neq s}} \left( \sum_{1 \leq k \leq R} P_{u_k} \mathbf{x}^d \right)^\top \Omega^{(d-j)} \mathbf{x}^j$$

$$= \sum_{1 \leq k \leq R} \left[ \mathbf{x}^{d\top} P_{u_k} \Omega^{(d-s)} \mathbf{x}^s - \frac{1}{d-1} \sum_{\substack{j \leq d \\ j \neq s}} \mathbf{x}^{d\top} P_{u_k} \Omega^{(d-j)} \mathbf{x}^j \right] \tag{25a}$$

$$c^{ds} = \mathbf{x}^{d\top} \Omega^{(d-s)} \left( \sum_{1 \leq k \leq R} P_{v_k} \mathbf{x}^s \right) - \frac{1}{d-1} \sum_{\substack{j \leq d \\ j \neq s}} \mathbf{x}^{d\top} \Omega^{(d-j)} \left( \sum_{1 \leq k \leq R} P_{v_k} \mathbf{x}^j \right)$$

$$= \sum_{1 \leq k \leq R} \left[ \mathbf{x}^{d\top} \Omega^{(d-s)} P_{v_k} \mathbf{x}^s - \frac{1}{d-1} \sum_{\substack{j \leq d \\ j \neq s}} \mathbf{x}^{d\top} \Omega^{(d-j)} P_{v_k} \mathbf{x}^j \right] \tag{25b}$$

When determining the subspaces corresponding to signals, we proceed with separate analyses for source and destination (similar to the process in Appendix B). We choose the smallest set of terms of (25a) summing to $c^{ds}$ to define $P_{\mathcal{U}}$, and the smallest set of terms of (25b) summing to $c^{ds}$ to define $P_{\mathcal{V}}$.

To perform communication tracing as described in §4 requires the corresponding adjustments to (29) and (30):

$$d^{ds}(\mathbf{w}) = \mathbf{w}^\top \Omega^{(d-s)} \mathbf{x}^s - \frac{1}{d-1} \sum_{\substack{j \leq d \\ j \neq s}} \mathbf{w}^\top \Omega^{(d-j)} \mathbf{x}^j \tag{26}$$

and

$$s^{ds}(W) = \mathbf{x}^{d\top}\Omega^{(d-s)}W_s - \frac{1}{d-1}\sum_{\substack{j \leq d \\ j \neq s}} \mathbf{x}^{d\top}\Omega^{(d-j)}W_j \tag{27}$$

after which the tracing procedure is unchanged.

## D  Proof of Lemma 1

Here we prove a generalization of Lemma 1.

**Lemma 1 (generalized).** Let $n \leq N$ be equal to the number of tokens considered in self-attention at head $(\ell, a)$ for destination token $d$, and $c^{\ell ads}$ be defined as in (2). For $0 < \alpha < 1$, if $A_{ds} \geq \alpha$, then $c^{\ell ads} > \sqrt{R}\left(\ln\left(\frac{\alpha}{1-\alpha}\right) + \ln(n-1)\right)$.

*Proof.* Assume $d > 1$. (If $d = 1$, the destination is the first token or the BOS token, which always places full attention only on itself.) By self-attention, we always have $s \leq d$. Accordingly $n$ is equal to $d$ (so by assumption, $n > 1$.)

Expanding the computation of attention weights from scores:

$$A_{ds} = \frac{\exp(A'_{ds}/\sqrt{R})}{\sum_j \exp(A'_{dj}/\sqrt{R})} = \frac{\exp(A'_{ds}/\sqrt{R})}{\sum_{j\neq s}\exp(A'_{dj}/\sqrt{R}) + \exp(A'_{ds}/\sqrt{R})}$$

So if $A_{ds} \geq \alpha$, then

$$\exp(A'_{ds}/\sqrt{R}) \geq \frac{\alpha}{1-\alpha}\sum_{j\neq s}\exp(A'_{dj}/\sqrt{R}).$$

By Jensen's inequality

$$\frac{1}{n-1}\sum_{j\neq s}\exp(A'_{dj}/\sqrt{R}) > \exp\left(\frac{1}{n-1}\sum_{j\neq s}A'_{dj}/\sqrt{R}\right),$$

so

$$\exp(A'_{ds}/\sqrt{R}) > \frac{\alpha}{1-\alpha}(n-1)\exp\left(\frac{1}{n-1}\sum_{j\neq s}A'_{dj}/\sqrt{R}\right)$$

so

$$A'_{ds}/\sqrt{R} > \ln\left(\frac{\alpha}{1-\alpha}\right) + \ln(n-1) + \frac{1}{n-1}\sum_{j\neq s}A'_{dj}/\sqrt{R}.$$

so

$$c^{\ell ads} > \sqrt{R}\left(\ln\left(\frac{\alpha}{1-\alpha}\right) + \ln(n-1)\right). \quad \square$$

So we see that a sufficient condition for $c^{\ell ads}$ to be positive is that $\ln(n-1) \geq -\ln\left(\frac{\alpha}{1-\alpha}\right)$. With positive $n$ and $\alpha$, this condition is met when $\alpha \geq 1/n$.

An important corollary is the contrapositive:

$$c^{\ell ads} < \sqrt{R} \left( \ln \left( \frac{\alpha}{1-\alpha} \right) + \ln(n-1) \right) \implies A_{ds} < \alpha.$$

Thus we can place an upper bound on the attention weight $\alpha$ as a function of $c^{\ell ads}$:

$$\exp(c^{\ell ads}/\sqrt{R})/(n-1) < \alpha/(1-\alpha) \implies A_{ds} < \alpha$$

or

$$\frac{\exp(c^{\ell ads}/\sqrt{R})}{n-1+\exp(c^{\ell ads}/\sqrt{R})} < \alpha \implies A_{ds} < \alpha \tag{28}$$

In §4, we consider the effect of upstream components on a downstream $c^{\ell ads}$. Here we note that (28) shows the precise nature of the nonlinear impact of upstream contributions on attention weight. The relationship (through the logistic) suggests that a more sophisticated approach (e.g., Shapley values) may be beneficial in determining the impact of each upstream component. We leave this extension for future work.

## E   Proof of Lemma 2

**Lemma 2.** Given vectors $\mathbf{x}$ and $\mathbf{y}$, among all rank-1 matrices having unit Frobenius norm, the matrix $D$ that maximizes $\mathbf{x}^\top D \mathbf{y}$ is $D = \frac{\mathbf{x}}{\|\mathbf{x}\|} \frac{\mathbf{y}^\top}{\|\mathbf{y}\|}$.

*Proof.* Given vectors $\mathbf{x}$ and $\mathbf{y}$, among all rank-1 matrices having unit Frobenius norm, the matrix $D$ that maximizes $\mathbf{x}^\top D \mathbf{y}$ is $D = \frac{\mathbf{x}}{\|\mathbf{x}\|} \frac{\mathbf{y}^\top}{\|\mathbf{y}\|}$.

First we show that any rank-1 matrix having unit Frobenius norm can be expressed as the outer product of two unit-norm vectors. Consider a rank-1 matrix $X$ having unit Frobenius norm. Since $X$ is rank-1, we can write $X = \mathbf{x}\mathbf{y}^\top$. Now construct $\tilde{X} = \frac{\mathbf{x}}{\|\mathbf{x}\|} \frac{\mathbf{y}^\top}{\|\mathbf{y}\|}$. By construction $\tilde{X}$ is both rank-1 and unit norm. Matrices $X$ and $\tilde{X}$ differ by a constant factor $\frac{1}{\|\mathbf{x}\|\|\mathbf{y}\|}$. However, since they have the same norm, we must have $\|\mathbf{x}\|\|\mathbf{y}\| = 1$, and so $X$ can be expressed as the outer product of two unit vectors.

Next consider a unit-norm, rank-1 matrix $G = \mathbf{u}\mathbf{v}^\top$ for unit vectors $\mathbf{u}$ and $\mathbf{v}$. By way of contradiction, suppose $\mathbf{x}^\top G \mathbf{y} > \mathbf{x}^\top D \mathbf{y}$. Then $\mathbf{x}^\top \mathbf{u}\mathbf{v}^\top \mathbf{y} > \mathbf{x}^\top \frac{\mathbf{x}}{\|\mathbf{x}\|} \frac{\mathbf{y}^\top}{\|\mathbf{y}\|} \mathbf{y}$. The right hand side is the positive quantity $\|\mathbf{x}\|\|\mathbf{y}\|$. The left hand side is the product of the projections of $\mathbf{x}$ onto $\mathbf{u}$, and $\mathbf{y}$ onto $\mathbf{v}$. The product is maximized when $\mathbf{u} = \mathbf{x}/\|\mathbf{x}\|, \mathbf{v} = \mathbf{y}/\|\mathbf{y}\|$, or $\mathbf{u} = -\mathbf{x}/\|\mathbf{x}\|, \mathbf{v} = -\mathbf{y}/\|\mathbf{y}\|$. In either case, $\mathbf{x}^\top G \mathbf{y} = \mathbf{x}^\top D \mathbf{y}$. □

## F   Details of the Experimental Setup

**Indirect Object Identification (IOI).**   The Indirect Object Identification (IOI) task [45] uses prompts that are structured as in this example: "When Mary and John went to the store, John gave the drink to". In this scenario, the model's goal is to predict *Mary*, which is the indirect object, rather than *John*. We used the authors' code to generate 256 prompts for this task, using a mix of the ABBA and BABA templates.

**Gender Pronoun (GP).**   In the Gender Pronoun (GP) task [60], the model is given prompts of the form: "So John is a really great friend, isn't". The objective for the model is to predict the correct pronoun, which would be either *he* or *she*. We used the authors' code with the 100 provided examples.

**Greater Than (GT).**   The Greater Than (GT) task, as described by [59], involves prompts of the form: "The attack lasted from the year 1920 to the year 19". In this task, the model's objective is to predict any two-digit number greater than 20. We utilized the authors' provided code to generate 256 prompts for this specific task. However, Gemma-2 2B was not analyzed for the GT task. The primary reason is that Gemma-2 2B's tokenizer predominantly uses single-digit tokens for numerical values,

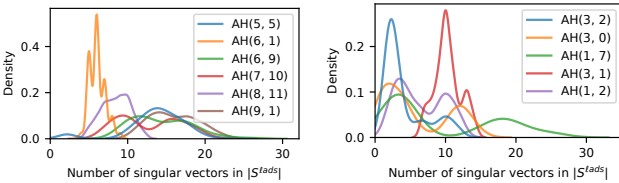

Figure 6: Dimension of signal subspace for key heads in the GT Task: (a) GPT-2 (b) Pythia.

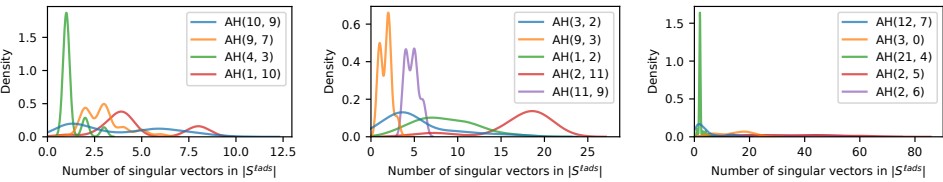

Figure 7: Dimension of signal subspace for key heads in the GP task: (a) GPT-2 (b) Pythia (c) Gemma-2.

which leads to two-digit numbers being a sequence of two separate tokens. Such a fundamental difference in number tokenization makes direct performance comparisons on this task challenging and potentially misleading.

**Handling Layer Norm** We use the TransformerLens library [64] for our experiments. To properly attribute contributions of upstream heads to downstream relative attention, we need to take into account the effect of the (downstream) layer norm. For GPT-2 and Pythia models, the layer norm operation can be decomposed into four steps: centering, normalizing, scaling, and translation. Gemma-2 models apply the same steps except for centering. Centering, scaling, and translation are affine maps, which means that they can be folded into different parts of the model with mathematical equivalence. The TransformerLens library handles the centering step by setting each weight matrix that writes into the residual stream to have zero mean. Moreover, it folds the scaling and translation operations into the weights of the next downstream layer.[5] The result is that centering, scaling, and translation make changes to the matrices used to compute $\Omega$ as shown in (1). The remaining step is the normalizing step. This step does not change the direction of the residual; it only affects the magnitude of the contribution to the relative attention (33). Since for any relative attention calculation, we are considering a specific addition to the residual $\mathbf{o}^i$, we can simply scale its contribution by the same scaling factor used for the corresponding token $\mathbf{x}^i$ when it is input to the downstream layer.

## G  Sparse Attention Decomposition

Here we provide additional examples showing that sparse attention decomposition holds across all tasks, models, and heads we examine. Figure 6 shows typical results for key heads in each model on the GT task, and Figure 7 shows typical results for key heads in each model on the GP task.

Figure 8 presents typical results for key attention heads in each model, specifically for the IOI task. For models like GPT-2 and Pythia, which have an $R$ value of 64, attention scores for various attention heads with distinct roles are typically computed using fewer than 20 dimensions, and sometimes even fewer than 5. A similar trend is observed in Gemma-2, which has a larger $R$ value of 256.

## H  Details of Tracing

Here we provide more detail and formal definitions to supplement §4. The functions $d^{\ell ads}(\mathbf{w})$ and $s^{\ell ads}(W)$ are defined as:

---

[5]See https://github.com/TransformerLensOrg/TransformerLens/blob/main/further_comments.md for more details.

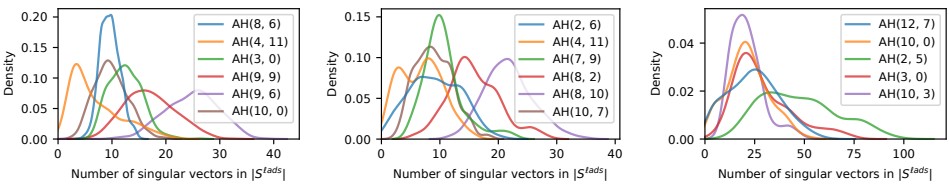

Figure 8: Dimension of signal subspace for heads in (a) GPT-2 (b) Pythia (c) Gemma-2, IOI task.

$$d^{\ell ads}(\mathbf{w}) = \mathbf{w}^\top \Omega^{\ell a} \mathbf{x}^{\ell s} - \frac{1}{d-1} \sum_{\substack{j \leq d \\ j \neq s}} \mathbf{w}^\top \Omega^{\ell a} \mathbf{x}^{\ell j} \tag{29}$$

where $\mathbf{w} \in \mathbb{R}^D$, and

$$s^{\ell ads}(W) = \mathbf{x}^{\ell d\top} \Omega^{\ell a} W_s - \frac{1}{d-1} \sum_{\substack{j \leq d \\ j \neq s}} \mathbf{x}^{\ell d\top} \Omega^{\ell a} W_j \tag{30}$$

where $W \in \mathbb{R}^{N \times D}$.

The computation of upstream contributions to relative attention relies on the linearity of $d^{\ell ads}(\mathbf{w})$ and $s^{\ell ads}(W)$ to distribute these functions over (6). Specifically, for any given contribution (characterizing an attention score) between tokens $d$ and $s$ at head $(\ell, a)$, we compute the portion of that contribution due to each upstream model component's writing into the destination token $d$ as

$$c^{\ell ads} = d^{\ell ads}(\mathbf{x}^{\ell d}) = \overbrace{\sum_{1 \leq l < \ell} \sum_{1 \leq h \leq H} \sum_{1 \leq t \leq N} d^{\ell ads}(\mathbf{o}^{lhdt})}^{\text{AH outputs}} + \overbrace{\sum_{1 \leq l < \ell} d^{\ell ads}(\mathbf{f}^{ld})}^{\text{FFN outputs}} + \overbrace{\sum_{1 \leq l < \ell} d^{\ell ads}(\mathbf{b}_O^l)}^{\text{attn. biases}} + \overbrace{d^{\ell ads}(\mathbf{x}^{0d})}^{\text{input at layer 0}} \tag{31}$$

and we write the portion of that contribution due to each upstream model component's writing into the source token $s$ as

$$c^{\ell ads} = s^{\ell ads}(X^\ell) = \overbrace{\sum_{1 \leq l < \ell} \sum_{1 \leq h \leq H} \sum_{1 \leq t \leq N} s^{\ell ads}(O^{lh*t})}^{\text{AH outputs}} + \overbrace{\sum_{1 \leq l < \ell} s^{\ell ads}(F^l)}^{\text{FFN outputs}} + \overbrace{\sum_{1 \leq l < \ell} s^{\ell ads}(\mathbf{1}\mathbf{b}_O^{l\top})}^{\text{attn. biases}} + \overbrace{s^{\ell ads}(X^0)}^{\text{input at layer 0}} \tag{32}$$

where $O^{lh*t} \in \mathbb{R}^{N \times D}$ is the matrix for which row $d$ is $\mathbf{o}^{lhdt}$ and $F^l \in \mathbb{R}^{N \times D}$ is the matrix for which row $d$ is $\mathbf{f}^{ld}$.

By (5), (7) and the properties of the SVD, we have

$$c^{\ell ads} \approx d^{\ell ads}(\mathbf{s}^{\ell d}) = d^{\ell ads}(P_{\mathcal{U}} \mathbf{x}^{\ell d}) \quad \text{and} \quad c^{\ell ads} \approx s^{\ell ads}(S^\ell) = s^{\ell ads}(X^\ell P_{\mathcal{V}}^\top). \tag{33}$$

The process of distributing (33) over (31) and (32) results in, for the destination token (repeating (9)):

$$c^{\ell ads} \approx \sum_{l < \ell, h \leq H, t \leq N} d_{lhdt}^{\ell ads} + \sum_{l < \ell} d_{ld}^{\ell ads} + \sum_{l < \ell} d_l^{\ell ads} + d_0^{\ell ads} \tag{34}$$

And for the source token is:

$$c^{\ell ads} \approx \sum_{l < \ell, h \leq H, t \leq N} s_{lhst}^{\ell ads} + \sum_{l < \ell} s_{ls}^{\ell ads} + \sum_{l < \ell} s_l^{\ell ads} + s_0^{\ell ads} \tag{35}$$

Where the terms in (34) and (35) are defined as:
$d_{lhdt}^{\ell ads} = d^{\ell ads}(P_{\mathcal{U}} \mathbf{o}^{lhdt}), \quad d_{ld}^{\ell ads} = d^{\ell ads}(P_{\mathcal{U}} \mathbf{f}^{ld}), \quad d_l^{\ell ads} = d^{\ell ads}(P_{\mathcal{U}} \mathbf{b}_O^l), \quad d_0^{\ell ads} = d^{\ell ads}(P_{\mathcal{U}} \mathbf{x}^{0d}),$
and
$s_{lhst}^{\ell ads} = s^{\ell ads}(O^{lh*t} P_{\mathcal{V}}^\top), \quad s_{ls}^{\ell ads} = s^{\ell ads}(F^l P_{\mathcal{V}}^\top), \quad s_l^{\ell ads} = s^{\ell ads}(\mathbf{1}\mathbf{b}_O^{l\top} P_{\mathcal{V}}^\top), \quad s_0^{\ell ads} = s^{\ell ads}(X^0 P_{\mathcal{V}}^\top),$
where $O^{lh*t}$ is the matrix for which row $d$ is $\mathbf{o}^{lhdt}$ and $F^l$ is the matrix for which row $d$ is $\mathbf{f}^{ld}$.

**Constructing Communication Graphs**  Communication graphs are constructed as follows. The objective is to identify the causal communication pathways within the model that lead to a particular output. We formalize this by first defining an "output of interest" based on the model's final logit predictions, $\mathbf{y}$. This output is a task-specific metric, calculated as a linear combination of logits, which can be represented as $\mathbf{a}^\top \mathbf{y}$. This formulation captures common task-specific success metrics.

- For the **Indirect Object Identification (IOI) task**, $\mathbf{a}^\top \mathbf{y}$ represents the logit difference between the indirect object (IO) and subject (S) tokens.
- For the **Gender Pronoun (GP) task**, it is the logit difference between the correct and incorrect gender pronouns (e.g., "he" versus "she").
- For the **Greater Than (GT) task**, it corresponds to the average logit difference between token predictions for two-digit numbers greater than a base number YY and those for numbers smaller than or equal to YY.

Using the vector $\mathbf{a}$ and the model's unembedding matrix $W_U \in \mathbb{R}^{D \times |\Sigma|}$ (where $|\Sigma|$ is the vocabulary size), we derive a "success direction" vector $\mathbf{g} = W_U \mathbf{a} \in \mathbb{R}^D$ in the residual stream space. For the IOI task, this would be $\mathbf{g} = W_U^{IO} - W_U^{S}$, where $W_U^t$ is the column of $W_U$ for token $t$. We then analyze the model's final residual stream output for the token position being predicted, which we denote $\mathbf{x}^{\text{out}}$. The projection of this vector onto the success direction, $\mathbf{x}^{\text{out}\top} \mathbf{g}$, quantifies the model's performance on the task for that specific prompt. We trace this value back by measuring the contribution of each component from the residual stream decomposition (Equation (6)) to this projection. "Seed" components are those with the largest contributions, specifically the smallest set of components whose contributions sum to the total projection value, $\mathbf{x}^{\text{out}\top} \mathbf{g}$. We also include any single component whose contribution is at least half of the total. If no attention heads are present among these seeds, tracing is not performed. The token at the output position serves as the initial destination token $d$ for tracing, while the source token for a seed attention head is identified from its OV circuit decomposition.

Starting from the identified seed components, we recursively trace their upstream contributors. Typically, the terms in equations (9) and (35) will show a few large values and many small values. As a result, it is important to filter these terms to isolate the important communication taking place in the model. To do so, we again rely on the property ensured by Lemma 1, namely, that $c^{\ell a d s}$ is a positive quantity. For each of (9) and (35), we select the smallest set of terms that sum to $\beta c^{\ell a d s}$ for some $0 < \beta \leq 1$. The parameter $\beta$ is set based on the degree to which the less important edges should be filtered from the communication graph. In practice, we use $\beta = 0.7$ in all our results, which filters most of the low-weight edges while preserving the largest-weight edges. Algorithm 1 outlines the pseudocode for the complete process of construction communication graphs.

# I   Communication Graphs

In this section we provide details on how we construct communication graph, and we show communication graphs for example prompts from the tasks and models we study.

We generated communication graphs and circuits for specific prompts that met two key criteria:

1. The model successfully predicted the correct answer for the prompt.
2. The prompt has at least one attention head in the "seeds" components set.

The total number of prompts that satisfied these conditions and were traced for each model and task is detailed in Table 1.

On CPU hardware (machines with 28 cores), tracing a 22-token prompt (the largest prompt size across all tasks that we used) takes approximately one minute for GPT-2 and for Pythia, and about one hour for Gemma-2. Our code is not highly optimized, and significant improvements are possible.

The time complexity for tracing a single prompt is $\mathcal{O}(a \cdot L \cdot C)$. In this expression, where $L$ is the number of layers, and $C$ denotes the number of components in the model, ie, attention heads, attention biases, MLPs, and embeddings (see Equation 6). The factor $a$ represents the number of significant attention values (i.e., those with $A_{ds}^{\ell a} \geq \frac{1}{n}$); it is upper-bounded by the number of attention values in a auto-regressive model $\frac{N^2 - N}{2} + N$, but is in practice much less because only heads

that make causal contributions downstream are included in the trace. Thus, model size differences explain the increased tracing time for Gemma-2 compared to GPT-2 and Pythia.

The following figures illustrate the communication graphs used by various models to solve example prompts from three distinct tasks:

- **Indirect Object Identification (IOI):** For the prompt *"Then, Simon and Andrew were working at the restaurant. Simon decided to give a basketball to"*, communication graphs are shown for GPT-2 small (Figure 9), Pythia-160M (Figure 10), and Gemma-2 2B (Figure 11).
- **Gender Pronoun (GP):** The communication graphs for GPT-2 small (Figure 12), Pythia-160M (Figure 13), and Gemma-2 2B (Figure 14) correspond to the prompt: *"So John is a really great friend, isn't"*.
- **Greater Than (GT):** For the prompt *"The consultation lasted from the year 1673 to the year 16"*, communication graphs are presented for GPT-2 small (Figure 15) and Pythia-160M (Figure 16).

## J   Interventions

An intervention targets an edge connecting an upstream (signal-writing) node and a downstream (signal-consuming) node. The intervention removes the signal from the downstream node's computation. Specifically, for a destination token edge, as characterized by Equation (9), the signal is removed from the *query* computation; for a source token edge, as characterized by Equation (35), the signal is removed from the *key* computation.

We implement two types of interventions: *boosting*, which involves adding the signal to the downstream attention head, and *suppressing*, which involves removing it. To evaluate these, we compare interventions using signals derived from $\mathcal{S}^{\ell ads}$ (i.e., the singular vectors identified by our method) against a random baseline, which is done by selecting $|\mathcal{S}^{\ell ads}|$ random singular vectors that are not in $\mathcal{S}^{\ell ads}$.

The effectiveness of an intervention is measured by the error $\frac{F(E,h)-F}{F}$, where $F(E,h)$ denotes the logit difference metric after the intervention, and $F$ represents this value before the intervention. A negative error value signifies that the model's performance has gotten worse, while a positive value indicates an improvement.

Figure 17 illustrates these intervention results across all three models and all three tasks. We observe that interventions using our signals are more causal than the random baseline. We further illustrate in Figure 18 that interventions have exactly the effect predicted by Lemma 1 – namely that signal ablation reduces attention and signal boosting increases attention. Finally, in Figure 19 we show the very small changes in vector norm and vector direction that result from these interventions.

In terms of time efficiency, performing an intervention requires approximately the same duration as a single forward pass through the model. To illustrate with our largest batch—consisting of 256 prompts, each 22 tokens long—executing an intervention on a 28-core CPU machine takes a few seconds for GPT-2 and Pythia, and around 2 minutes for Gemma-2.

## K   Circuits

Leveraging the observation that interventions can causally affect model performance, we use this insight as a basis for identifying circuits. Our method for finding circuits from communication graphs begins by aggregating multiple such graphs into a single graph, $G$. From this aggregated graph, we first prune nodes that appear in less than 1% of the instances. Subsequently, edges are removed based on their impact on downstream task performance. Specifically, an edge $E$ that occurred in $p$ prompts is removed from $G$ if its average intervention impact—calculated as $\frac{1}{p}\sum_i \left|\frac{F_i(E,h)-F_i}{F_i}\right|$— falls below a threshold $T$. In this formula, $F_i$ and $F_i(E,h)$ represent the logit difference metrics before and after the intervention for prompt $i$, respectively, and $T$ is a threshold determined empirically for each task and model. Figures 20, 21, and 22 shows the precision, recall, and $F_1$-scores, respectively, considering every possible method as a baseline for comparison.

## L  Handling logit soft-capping in Gemma-2 models

Gemma-2 models have soft-capping for attention scores and logits. The soft-capping is given by

$$f(x) = c \cdot \tanh\left(\frac{x}{c}\right),$$

where $c$ is the soft-capping constant ($c = 50$ for attention scores and $c = 30$ for the final logits in Gemma-2 [55]) and $x$ are either the scores of the logits. To handle non-linearity in our tracing, we used the first non-zero term of the Taylor expansion for the logit soft-capping function, yielding the approximation $f(x) \approx x$ (valid when $\frac{x}{c}$ is small). The Gemma 2 team observed very minor differences when soft-capping is removed during inference [65]. Moreover, we saw empirically that $f(x) \approx x$ is a good enough approximation for the soft-capping. Figure 23 shows the attention scores and their respective values after soft-capping for an IOI prompt.

## M  Control Signals

Figures 24 and 25 illustrate how control signals vary across the layers of the models we study. In these figures, the colors used for signals correspond to those in Figure 5. These control signals were initially identified using the IOI task with a subset of the prompts. To verify their broader applicability, Figures 24 and 25 were specifically generated using a prompt not part of this initial set. Other prompts that are not in this initial set also have very similar behavior. The consistent appearance of these signals on prompts that were not used to find the control signals corroborates their *data-independent* nature.

We also show in Figure 26 the distribution of inner products between the $V$ signals and zero and non-zero token signals for GPT-2, Pythia-160M, and Gemma-2 2B. We observe that the $V$ signals have considerably higher inner products with zero token signals than non-zero token signals across all the models.

**Algorithm 1:** Communication Graph Construction

*// Main function to find seeds and initiate tracing*

1   `ConstructGraph`$(M, P, \mathbf{a}, \beta)$

    **Input** : $M$: The transformer model.

           $P$: The input prompt.

           $\mathbf{a}$: A vector defining the task metric over the logit vocabulary.

           $\beta$: The contribution threshold for filtering ($0 < \beta \le 1$).

    **Output:** $G$: The final communication graph.

2      $G \leftarrow$ InitializeEmptyGraph();

3      Activations, $\mathbf{y} \leftarrow$ ForwardPass($M, P$);

4      $W_U \leftarrow$ GetUnembeddingMatrix($M$);

5      $\mathbf{g} \leftarrow W_U \mathbf{a} \triangleright$ *Calculate the "success direction" vector*

6      $\mathbf{x}^{out} \leftarrow$ GetFinalResidual($P$, Activations);

7      PerformanceScore $\leftarrow \mathbf{x}_{out}^{\top} \mathbf{g}$;

    *// Decompose the final residual and project contributions onto the success direction*

8      Contributions $\leftarrow$ DecomposeResidual($\mathbf{x}^{out}$) $\triangleright$ *Using Eq. (6)*

9      ProjectedContributions $\leftarrow \{c_i^{\top} \mathbf{g}$ for each $c_i \in$ Contributions$\}$;

    *// Identify the smallest set of components whose scores sum to the total score*

10      $S \leftarrow$ FindSmallestSubsetSum(ProjectedContributions, PerformanceScore);

    *// Also include any single component that contributes more than half the total score*

11      $S_{high} \leftarrow \{c_j \mid c_j^{\top} \mathbf{g} \ge 0.5 \times$ PerformanceScore$\}$;

12      Seeds $\leftarrow S \cup S_{high}$;

13      **foreach** *seed component* $(\ell, a, d, s) \in$ *Seeds* **do**

14          **if** $(\ell, a)$ *is an attention head* **then**

15              `RecursiveTrace`$(G, \ell, a, d, s, \beta)$;

16      **return** $G$;

17 ────────────────────────────────────────

*// Recursive helper function to trace upstream contributors*

18   `RecursiveTrace`$(G, \ell, a, d, s, \beta)$

    **Input** : $G$: The graph (modified in-place).

           $(\ell, a, d, s)$: The current component (layer, head, destination token, source token).

           $\beta$: The contribution threshold. We use $0.7$ in the experiments.

    **Output:** Modifies the graph $G$.

    $\triangleright$ *Base Cases: Stop recursion for all these cases or for when attention weight less than $1/n$ (see Lemma 1)*

19      **if** $\ell = 0$ *or* $d = 0$ *or* $d < s$ *or* $A_{ds}^{\ell a} < 1/n$ **then**

20          **return**;

21      $c^{\ell ads} \leftarrow$ CalculateRelativeAttention($\ell, a, d, s$);

    *// Find upstream contributors to the destination token's signal*

22      $U(d) \leftarrow$ FindUpstreamContributors($d, c^{\ell ads}, \beta$) $\triangleright$ *Using Eq. (9)*

23      **foreach** *upstream component* $(\ell', h', d, t) \in U(d)$ **do**

24          Add edge $(\ell', h', d, t) \rightarrow (\ell, a, d, s)$ to $G$;

25          `RecursiveTrace`$(G, \ell', h', d, t, \beta)$;

    *// Find upstream contributors to the source token's signal*

26      $U(s) \leftarrow$ FindUpstreamContributors($s, c^{\ell ads}, \beta$) $\triangleright$ *Using Eq. (35)*

27      **foreach** *upstream component* $(\ell', h', s, t) \in U(s)$ **do**

28          Add edge $(\ell', h', s, t) \rightarrow (\ell, a, d, s)$ to $G$;

29          `RecursiveTrace`$(G, \ell', h', s, t, \beta)$ $\triangleright$ *Source token is destination upstream.*

Table 1: Number of Traced Prompts per Model and Task

| Model | IOI | GT | GP |
|-------|-----|-----|-----|
| GPT-2 Small | 230 | 166 | 100 |
| Pythia-160M | 159 | 39 | 99 |
| Gemma-2 2B | 206 | – | 94 |

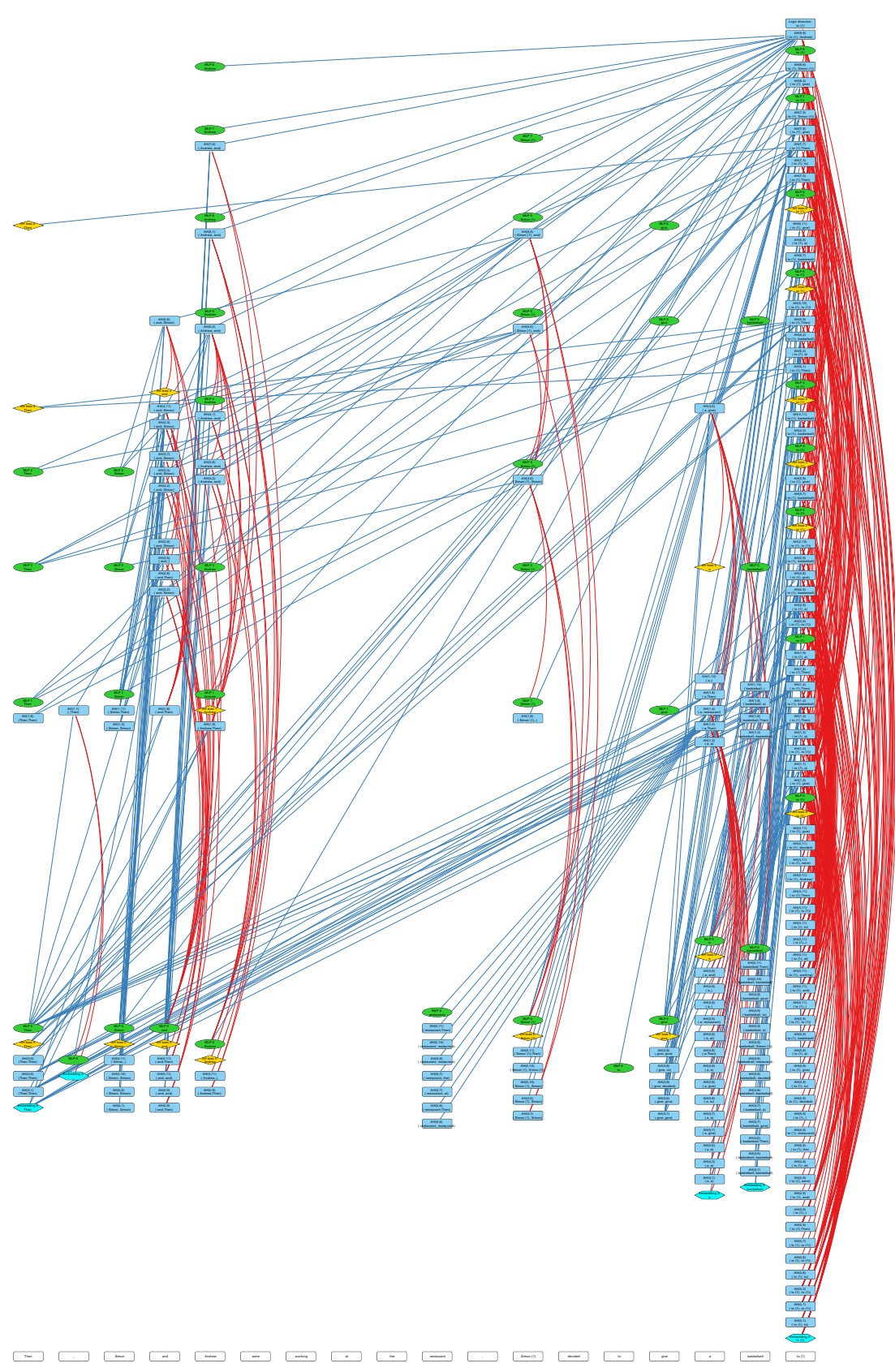

Figure 9: Communication graph used for GPT-2 small to solve an IOI prompt, with 247 nodes and 683 edges. The prompt used is: *"Then, Simon and Andrew were working at the restaurant. Simon decided to give a basketball to"*.

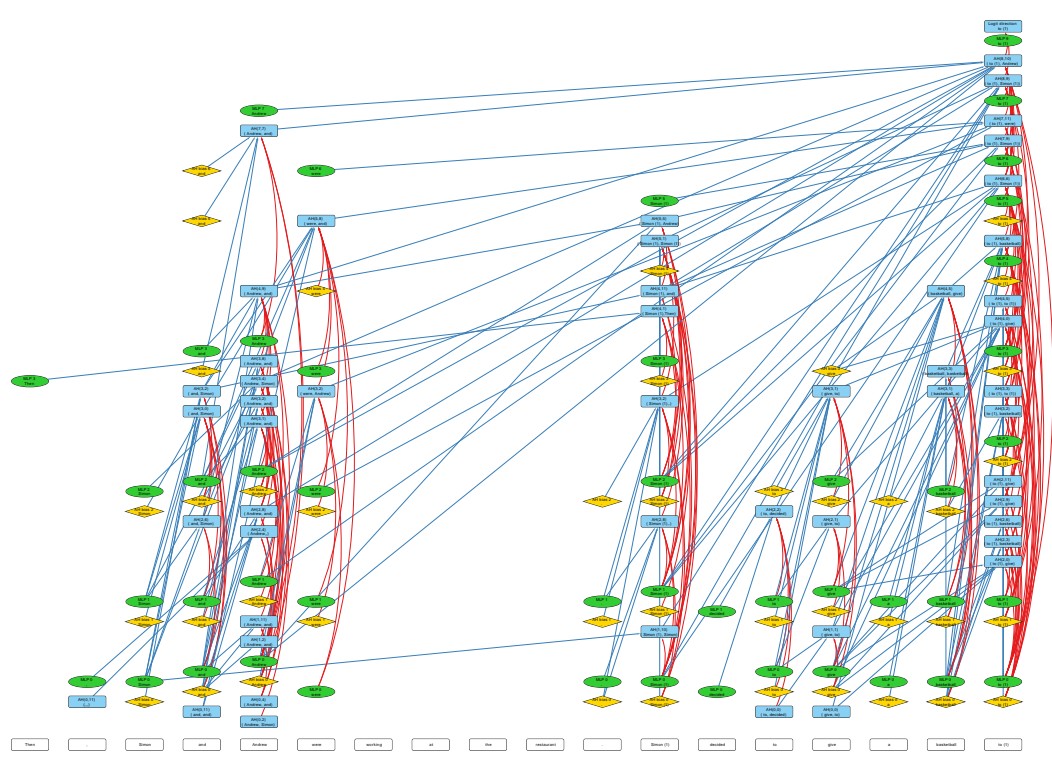

Figure 10: Communication graph used for Pythia-160M to solve an IOI prompt, with 158 nodes and 344 edges. The prompt used is: *"Then, Simon and Andrew were working at the restaurant. Simon decided to give a basketball to"*.

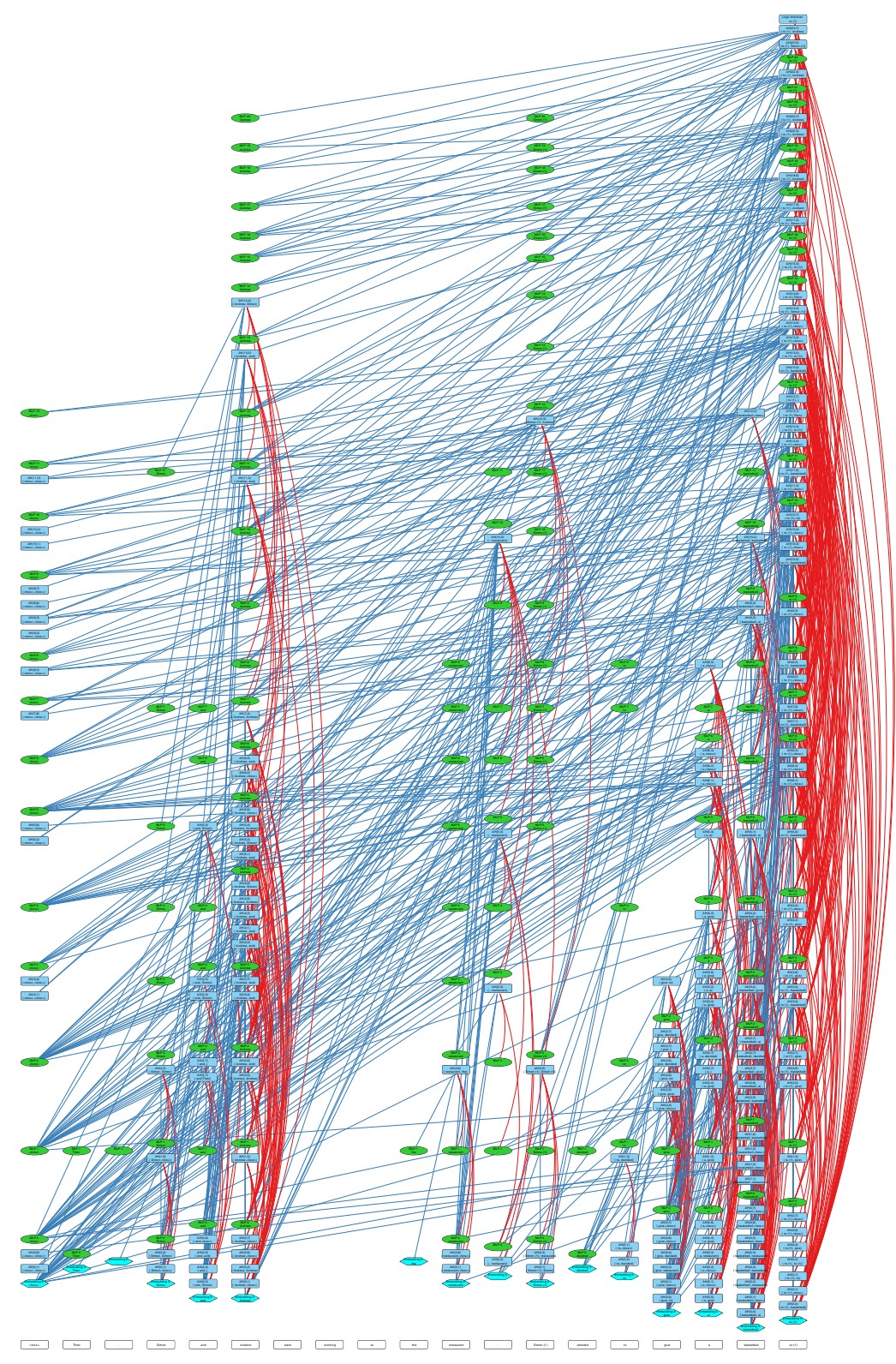

Figure 11: Communication graph used for Gemma-2 2B to solve an IOI prompt, with 352 nodes and 1358 edges. The prompt used is: *"Then, Simon and Andrew were working at the restaurant. Simon decided to give a basketball to"*.

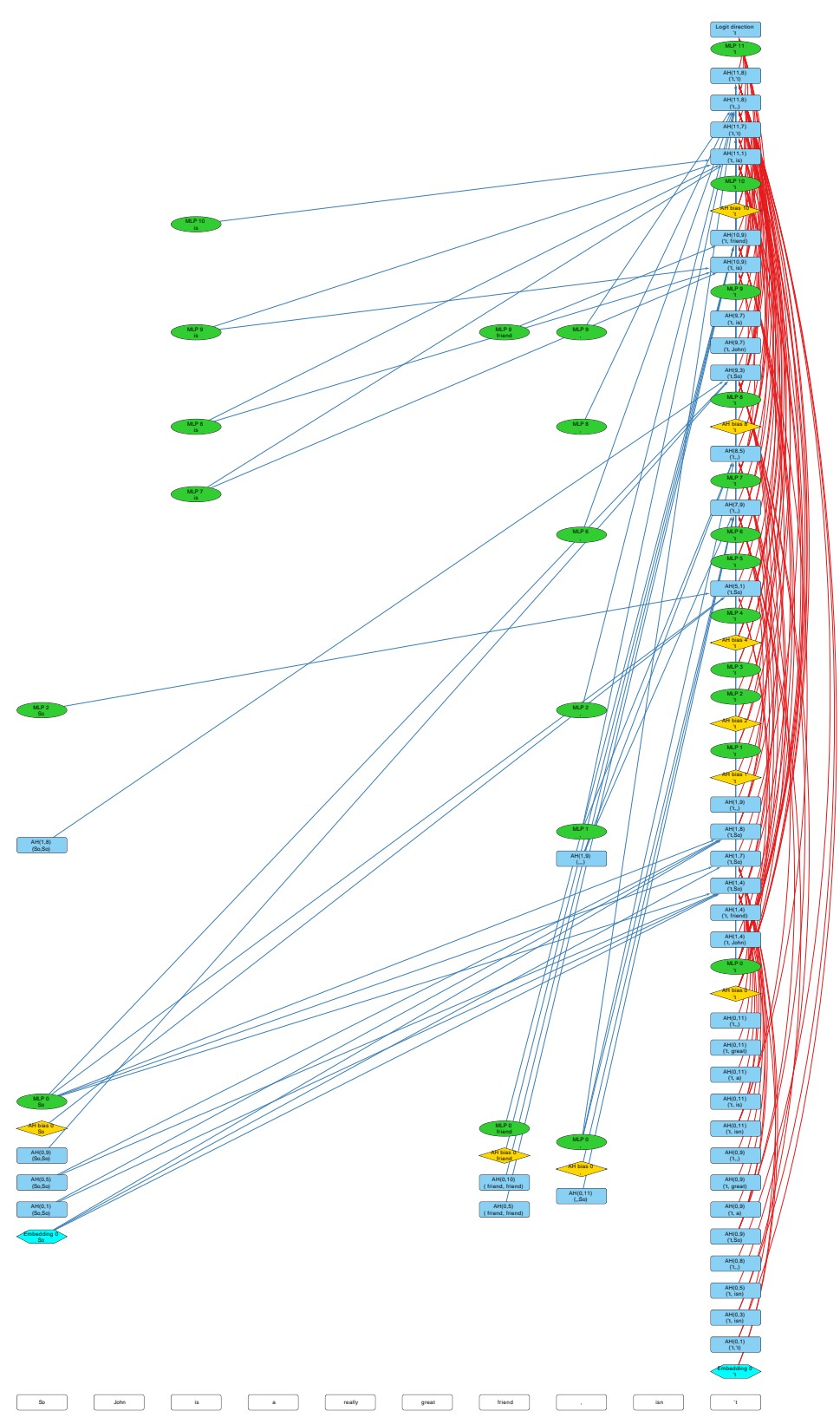

Figure 12: Communication graph used for GPT-2 small to solve a GP prompt, with 87 nodes and 152 edges. The prompt used is: *"So John is a really great friend, isn't"*.

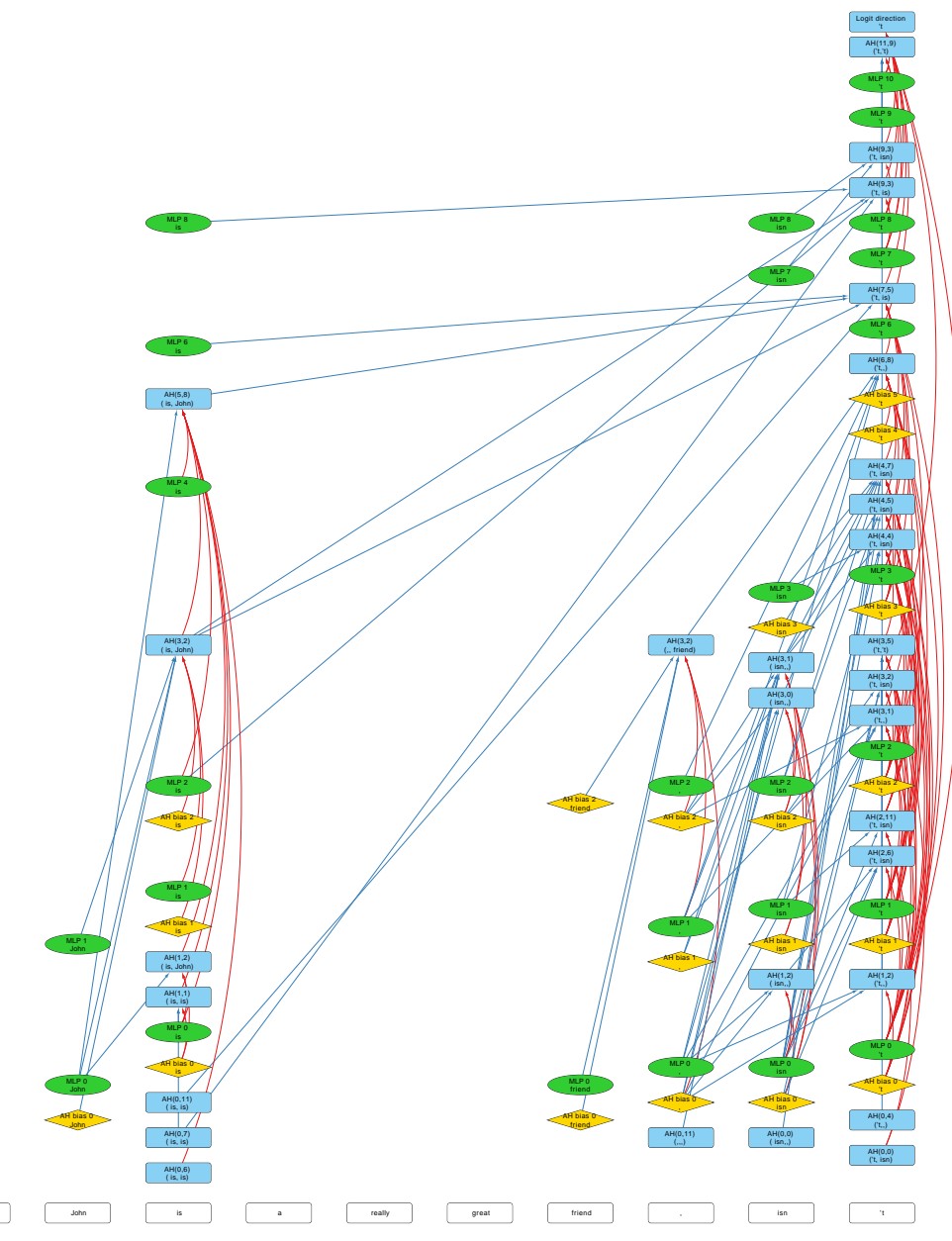

Figure 13: Communication graph used for Pythia-160M to solve a GP prompt, with 86 nodes and 172 edges. The prompt used is: *"So John is a really great friend, isn't"*.

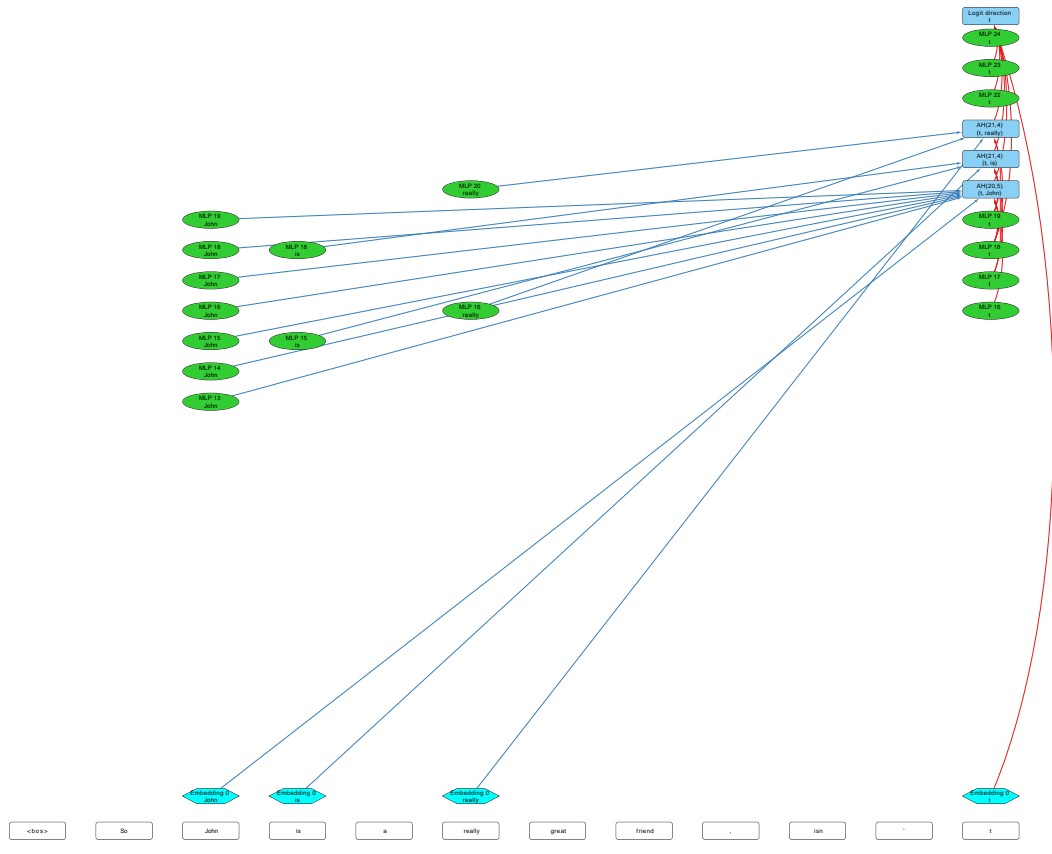

Figure 14: Communication graph used for Gemma-2 2B to solve a GP prompt, with 38 nodes and 31 edges. The prompt used is: *"So John is a really great friend, isn't"*.

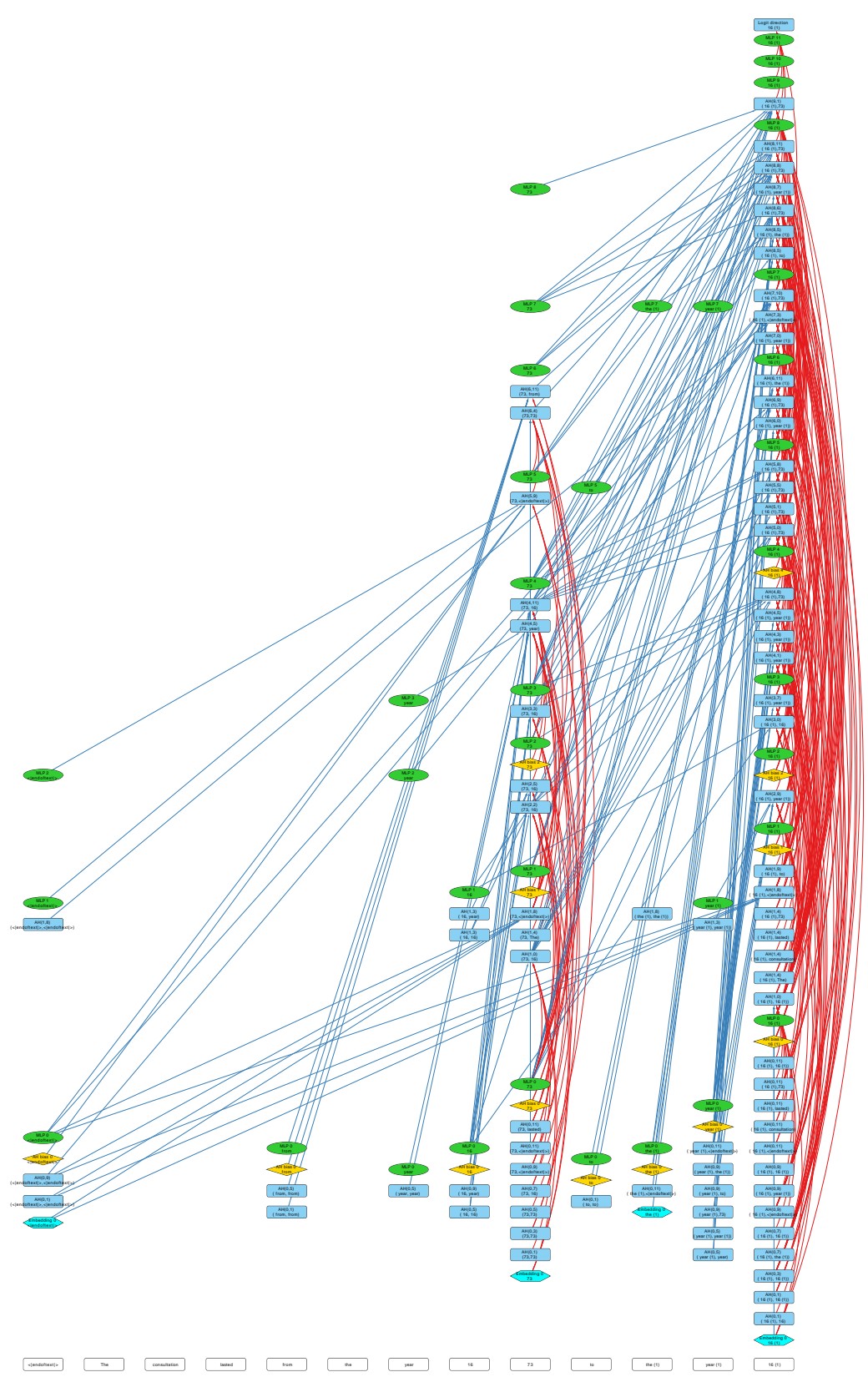

Figure 15: Communication graph used for GPT-2 small to solve a GT prompt, with 150 nodes and 412 edges. The prompt used is: *"The consultation lasted from the year 1673 to the year 16"*.

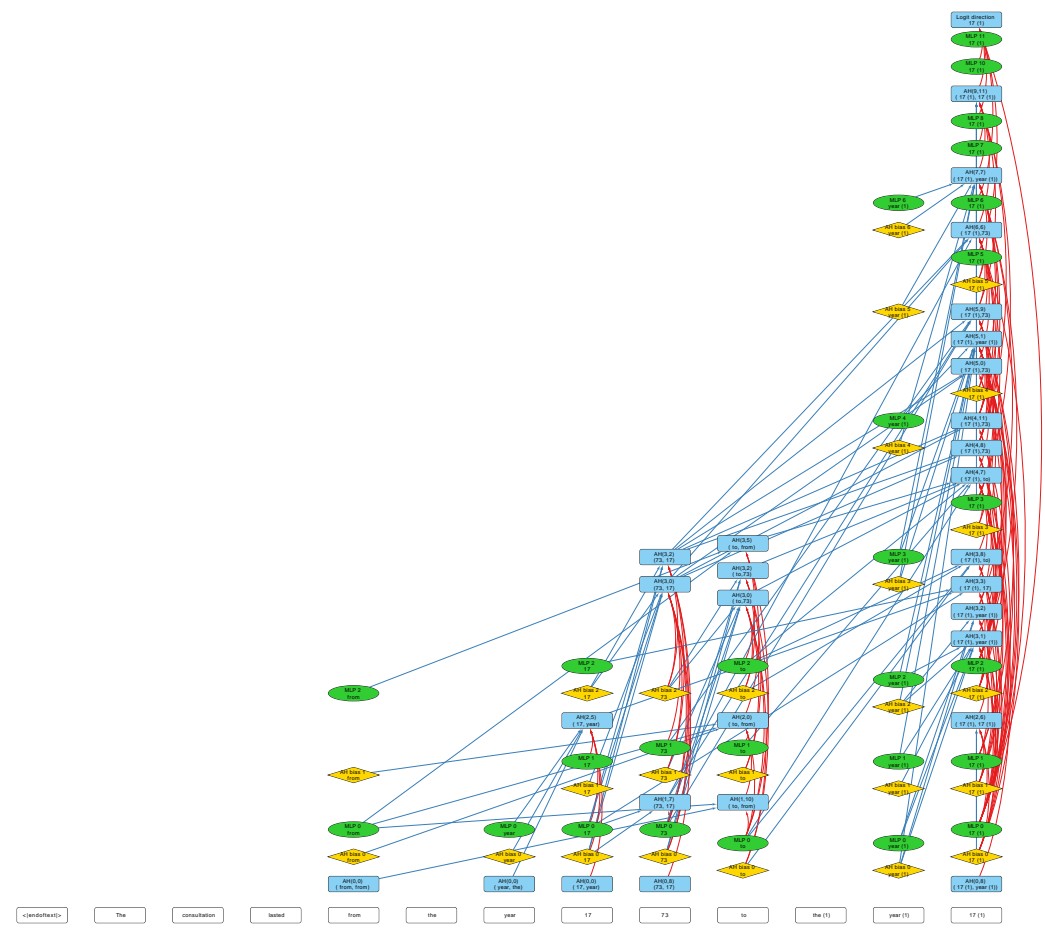

Figure 16: Communication graph used for Pythia-160M to solve a GT prompt, with 94 nodes and 189 edges. The prompt used is: *"The consultation lasted from the year 1673 to the year 16"*.

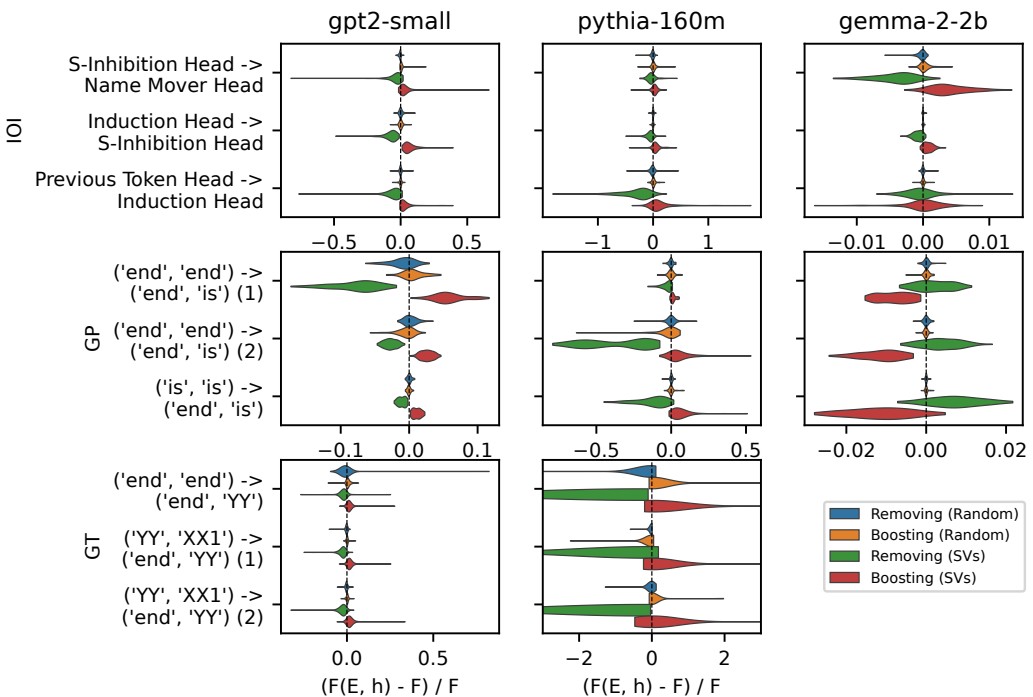

Figure 17: Intervention effect on GPT-2, Pythia, and Gemma-2 in the IOI, GP, and GT tasks. Green: signal ablation; Red: signal boosting; Blue: random ablation; Orange: random boosting.

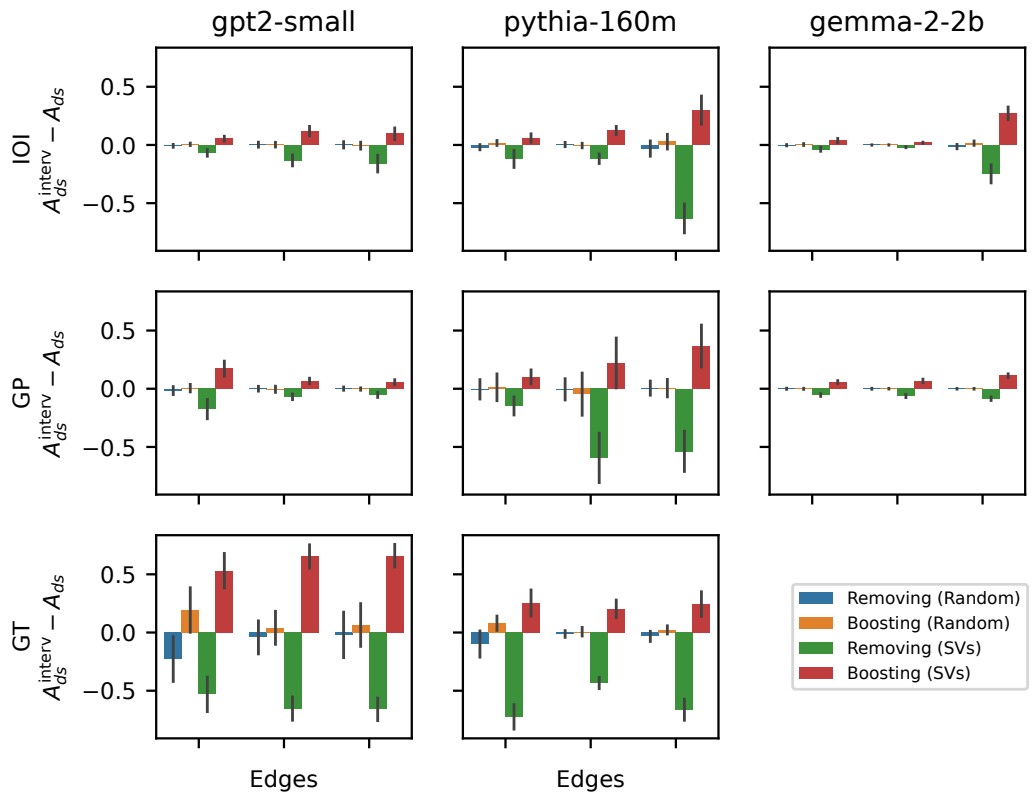

Figure 18: Interventions effect on the attention weight. Error bars are the standard deviation.

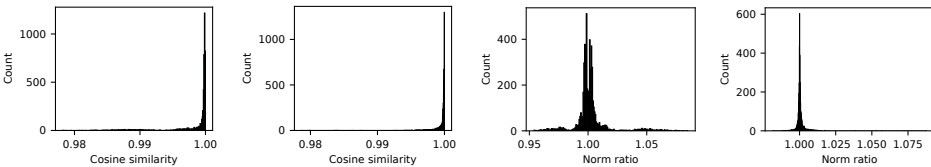

Figure 19: Distribution of cosine similarities and norm ratios between the intervened input residual and the original input residual (a) SVs (b) Random (c) SVs (d) Random.

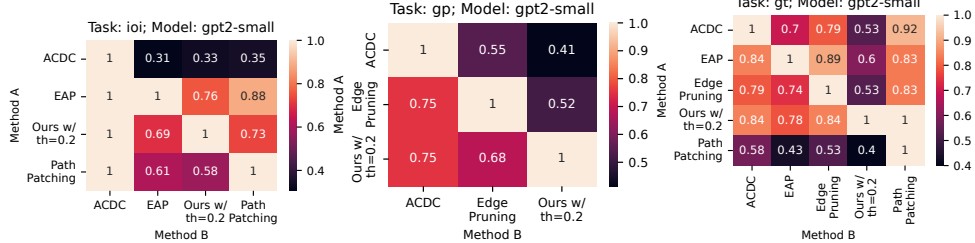

Figure 20: Heatmap with Precision scores considering Method A as a baseline for comparison for (a) IOI task (b) GP task (c) GT task.

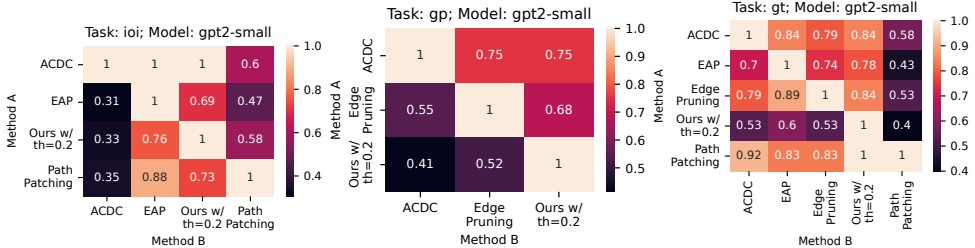

Figure 21: Heatmap with Recall scores considering Method A as a baseline for comparison for (a) IOI task (b) GP task (c) GT task.

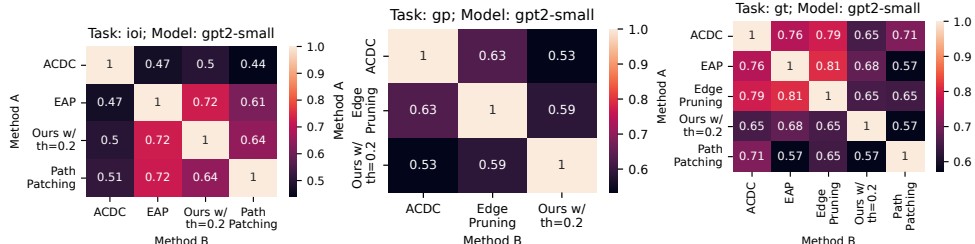

Figure 22: Heatmap with $F$-scores considering Method A as a baseline for comparison (a) IOI task (b) GP task (c) GT task.

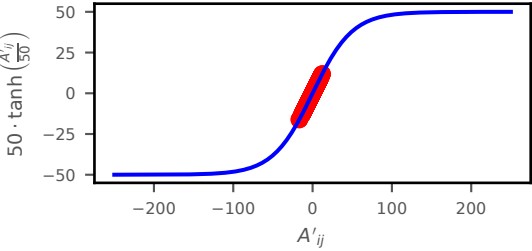

Figure 23: The attention scores and their respective values after soft-capping for an IOI prompt.

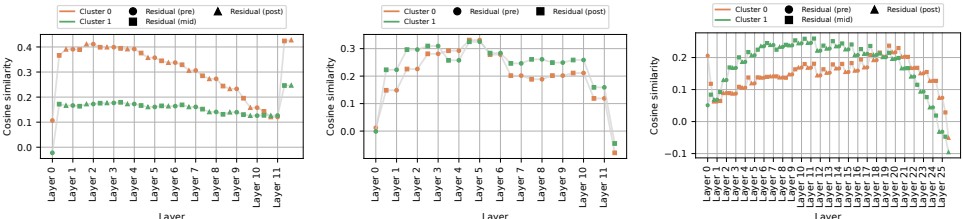

Figure 24: Cosine similarity between destination control signals and the residuals of the last token. (a) GPT-2 (b) Pythia (c) Gemma-2.

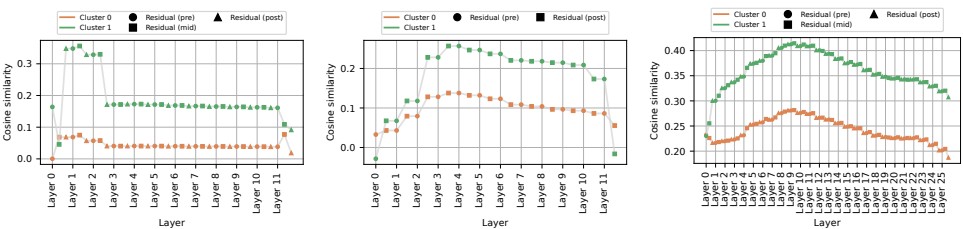

Figure 25: Cosine similarity between source control signals and the residuals of the first token. (a) GPT-2 (b) Pythia (c) Gemma-2.

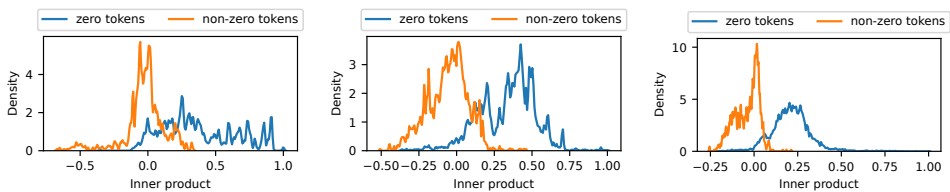

Figure 26: Distribution of inner products between the $V$ signals and zero and non-zero token signals in (a) GPT-2 (b) Pythia (c) Gemma-2.

