# OpenReview forum: "Pinpointing Attention-Causal Communication in Language Models"
_NeurIPS.cc/2025/Conference — NeurIPS 2025 poster_

### Official Review · Reviewer_tbcy · 2025-06-07

**Clarity:** 4
**Significance:** 3
**Originality:** 4
**Rating:** 5
**Confidence:** 5

**Summary:**

The work studies LLMs attention mechanism, by identifying low-dimensional features that enable the activation of individual attention heads, by identifying subspaces that the QK (query-key) matrices read from. Naming this features as "signals", the paper show how it can by applied to circuit discovery. In an experimental setup of GPT2 and Pythia, the paper show how this method can achieve similar circuit discovery performances to methods such as ACDC and EAP. One of the model's promises is the ability to mechanistically interpret models' attention-heads' behavior, such as attention sink, by identifying the features that enable them.

**Questions:**

Notes:
1. some works are cited twice in the References, like [49] and [58].
2. rearrange figures order: since figure 4 is presented in the text before figure 3, the index of the two should be replaced.
3. Regarding Figure 6 and its corresponding discussion: I would really appreciate more conversation on why should the model use global signals patterns. Although the results are mostly solid, it is counter intuitive to believe that only a small number of subspaces (features) dominates most of attention head.
4. Can you draw a connection between some signals and intermediate layer prediction (such as early exit projection [3][4])? I wonder if the signals has a meaning only when read by attention heads, or, that they hold some interpretable semantic.
[3] interpreting GPT: the logit lens

[4] A Mathematical Framework for Transformer Circuits

**Ethical Concerns:**

["NO or VERY MINOR ethics concerns only"]

**Final Justification:**

My main concerns were with the presentation and additional discussions which I think can make the work more comprehensive. The authors address these concerns in their official response. Overall, I found the work to be solid and innovative.

**Limitations:**

yes

**Quality:**

3

**Strengths And Weaknesses:**

Strengths:
1. The paper is well written and easy to follow for researcher from the field of mechanistic interpretability.
2. The method is highly based on model mechanistic and mathematical approximation - making the method highly justify.
3. The motivation, as well as the method itself, are very well justified. The results of section 5 also highlights the method power.

Weaknesses:
1. Unclear figures: Figure 4 is hard to read as most lines are over each others. Figure 11-17 are also hard to follow but I assume they are proposed to be view via a separate pdf file.
2. Regarding Figure 6 and its corresponding discussion: Since LLMs hidden state suffer from rouge dimensions [1][2] it is possible that the clustering be improved via another method than naive cosine-similarity, such as Spearman correlation.

[1] All Bark and No Bite: Rogue Dimensions in Transformer Language Models Obscure Representational Quality

[2] Outlier Dimensions that Disrupt Transformers are Driven by Frequency

---

> ### Author Rebuttal · Authors · 2025-07-30
>
> We thank the reviewer for their helpful review of our paper.
>
> ### Weaknesses:
> 1. Thank you for your suggestion about Figure 4. There is some overplotting in these figures, and we will improve them. Regarding Figures 11-17, indeed, they are proposed to be viewed in a separate file. We plan to release a Cytoscape file with Figures 11-17, which will enable them to be visualized interactively.
> 2. This is a very good point, thank you for pointing it out. We did not think about it, and indeed, using a rank correlation such as Spearman correlation, could improve the clustering. We will definitely investigate that in the revision.
>
> ### Questions:
> 1. Good catch; we will fix those in the revision.
> 2. Also helpful and we will fix the figure order in the revision.
> 3. Thank you for the question. Indeed, it is quite surprising that a few signals are responsible for so many attention weights.
> To explain, we start by noting that what is happening in all these cases is attention sink – meaning that the head is attending to the first token (or punctuation token) because it does not have any important function to perform.  In fact, for most of the attention heads and token pairs,  the model only performs the attention sink operation.  In other words, only a small subset of heads and token pairs are actually important for computing the model output in these tasks.  This is perhaps the underlying cause of the effect.
> We connect this fact with the observation (from [19]) that when models represent features in superposition, they will tend to allocate a dedicated dimension for features that are used frequently. Because attention sink is so common, the model seems to allocate a dedicated, global feature direction to control this behavior. The simplicity of this function, which is merely to direct attention to the first token, could also explain why a single direction suffices.
> 4. We have preliminary evidence that some of the signals our method identifies carry semantics that are shared across different attention heads. For example, we have found that in the IOI task [42], different Duplicate Token Heads and Induction Heads write the *exact same signal* (same one-dimensional subspace) to be read by S-inhibition heads. Another example is the control signals we identified, which appear to represent consistent, meaningful instructions that are shared across multiple heads and layers. However, the extent to which signals in general are shared remains an open question. A more in-depth study of their semantics is an exciting direction for future work that we are eager to pursue.

---

> > ### Comment · Reviewer_tbcy · 2025-08-02
> >
> > Thanks for the detailed answer. I found it to strength the score I gave the work and I hope to see its suggested fix in future versions.

---

### Official Review · Reviewer_kEJw · 2025-06-25

**Clarity:** 3
**Significance:** 3
**Originality:** 3
**Rating:** 5
**Confidence:** 3

**Summary:**

This paper proposes an analysis of model weights and representations to decompose attention computations into low-rank subspaces. They show that low-rank QK signals causally influence when a token is attended to, and use their decomposition to perform circuit discovery and analyze attention sink behavior in LLMs.

**Questions:**

- The results in Section 3 were a little unclear to me. Can you explain how you get distributions for the $\mathcal{S}^{\ell ads}$ plots in Figure 2? Is it just because it’s over many attention heads? Or does this also take into account many prompts, and within prompts many (all?) token pairs? Or only for specific token pairs in a given prompt?
- In Figure 18 (appendix J), the “boosting” vs. “removing” signals for gemma-2 seem to follow an opposite trend for the GP task. Any idea what’s going on there?
- Can you clarify the use of counterfactual inputs /counterfactual token pairs in your circuit-finding adaptation of your method? (e.g., line 765 describes using “he” vs “she” in the GP setting, and line 770 seems to describe taking a difference between embeddings of counterfactual tokens in the IOI setup)
- What are your thoughts on related work studying "binding" in LLMs? (e.g., [1-4]). These works also identify low-rank subspaces that seem to control different attributes of or attention to entities in a context. It seems related to the idea you're studying of how attention-causal mechanisms work, but did not see a discussion and am curious if you think there's any overlap between those findings and your  own.

___
[1] Feng and Steinhardt. [How do Language Models Bind Entities in Context?](https://openreview.net/pdf?id=zb3b6oKO77)

[2] Dai et al. [Representational Analysis of Binding in Language Models](https://aclanthology.org/2024.emnlp-main.967.pdf)

[3] Prakash et al. [Fine-Tuning Enhances Existing Mechanisms: A Case Study on Entity Tracking](https://openreview.net/pdf?id=8sKcAWOf2D)

[4] Prakash et al. [Language Models use Lookbacks to Track Beliefs](https://arxiv.org/pdf/2505.14685)

**Ethical Concerns:**

["NO or VERY MINOR ethics concerns only"]

**Final Justification:**

The rebuttal has clarified my concerns related to the use of the phrase "counterfactual", and clarified minor inconsistencies or details that were not included in the initial submission.

I have also read through the other reviewers' responses and rebuttals and I think the proposed changes will be helpful for increasing the clarity of the paper. With the rebuttal response addressing my concerns, I am inclined to Accept the paper.

**Limitations:**

Yes

**Quality:**

4

**Strengths And Weaknesses:**

**Strengths:**
- The paper is well written and fairly straightforward to follow. Several times I found myself having a question while reading, which was then almost immediately answered or had a reference to an appendix section with more detail.
- The authors are aware of relevant related work, and the paper’s contributions are contextualized nicely.
- The notation being consistent was very helpful when trying to understand the definitions in the presentation of the method.
Understanding attention-causal behavior is an interesting alternative to other circuit-finding work, and has promise if it does not rely on counterfactual pairs.

___
**Weaknesses:**
- The paper is pretty dense, and because of this many details of results are left to the appendix. This made some results a little unclear when presented in the main paper. For example, in Figure 2 (Section 3), the distribution is taken over an unspecified set of data (e.g. Is it just attention heads for specific token pairs, or all token pairs, all attention heads across many prompts?).
- In the paper you say your method does not depend on counterfactual inputs (e.g. lines 9-10, line 61), but you often describe parts of the method being reliant on what seem to be counterfactual token pairs (e.g. lines 249-251). I think I may have misunderstood something about this part of the definition - can you clarify this, and whether you’re using counterfactual token pairs here? (See also the question below)
- There were two terms I was unfamiliar with that it might be nice to have a citation or clarification for. These were the “Courant-Fischer theorem” (line 49, 188), and an “ersatz function” (line 149, 369).
- It's unclear whether the method feasibly scales to larger parameter models, but this is a minor limitation. They have shown it works fairly well for up to 2B models.

___
**Other:**
Minor Typos:
- Line 184: Duplicated citation for Pan et al. (uses both [49] & [58])
- Line 295: “postive” -> positive?

---

> ### Author Rebuttal · Authors · 2025-07-30
>
> We thank the reviewer for their helpful review of our paper.
>
> ### Weaknesses:
>
> - Thank you for pointing out this lack of clarity. These plots are over all the prompts, and over all the attention heads and respective token pairs (d, s) that appeared in a trace from our tracing algorithm. We see that our use of the term “relevant” didn’t clearly specify what they are. We will change the manuscript to make this part clearer.
> - We can see that there are multiple kinds of counterfactuals that can get confused here.  We discuss this bullet point and how we will clarify the issues as part of our response to the third “Questions” bullet below.
> - Thank you for pointing this out. We will add a citation for the Courant-Fischer theorem. Regarding the “ersatz function”, another term would be “replacement function”. We can see that that term would be a bit clearer so we’ll make that change in the revision.
> - Yes, the method scales well. All SVDs for a model can be computed just once and reused for every prompt. The bottleneck of this method is the same as the attention mechanism, which is quadratic in the number of tokens. We have a small discussion about this in the Appendix (lines 802-808).
>
> ### Questions:
> - We address the first question in the first bullet under Weaknesses above.
> - Yes, something odd is going on in that case. What is definite is that our method is causal in the attention weights – this is a provable fact. In other words, if we intervene in the position (d, s), the boosting and removing interventions increase and decrease the attention weight in the (d, s) position, respectively.
> So what we conclude for this case is that increasing the attention weight on the token pair in the downstream head actually decreases the performance of the model. We did not investigate further, but one possible explanation is that what this attention head in the (d, s) position is adding to the residual stream is information about suppressing the correct prediction (something similar to the Negative Name Mover heads in the IOI paper). Therefore, boosting this signal will make the performance worse, while ablating it will make the performance better.
> - Thanks for pointing out the need for clarification regarding “counterfactual inputs” along with the comparisons being made between alternatives at lines 249-251 and lines 759-771.
>
> In the style of circuit tracing that is most common in the literature (causal mediation analysis) a **counterfactual input** is a prompt that is identical to an original input, except that it differs in the single feature under study. For example, the paper [42] – which was the first to trace the IOI circuit – uses counterfactual inputs that vary a name across three different versions of a prompt. Inputs, therefore, come in pairs—the original and the counterfactual—and these pairs form the basis of their causal mediation analysis method. For instance, for the IOI task, the original and counterfactual inputs could be:
> - **Original input:** “When Mary and John went to the store, John gave a drink to”.
> - **Counterfactual input:** “When Mary and John went to the store, Ryan gave a drink to”.
>
> When we state that our method does not require “counterfactual inputs” (e.g., at lines 9-10), we are referring to this specific type of paired, minimally-different prompt As we note, constructing counterfactuals to test hypotheses about circuits can in many cases be a subtle and challenging task (eg, see [46]).
>
> We also use the term “counterfactual” in a different context, in which we are discussing a hypothetical situation in which a particular token representation could have been different.   This is the sense on lines 26, 33, and 42 and particularly on line 141.  The intent in those passages is to highlight that the signals we identify are causal for a head’s attention.   For example, at line 141 we are establishing causality by showing that, had a signal not been present, the head would not have attended to the token pair.  These passages are not considering actual counterfactual token pairs, but rather imagined modifications of a single token.
>
> Lines 249-251 are expanding on this hypothetical situation.  These lines are providing a mathematical tool for assessing the impact on attention if a token had been different.  So here, we are again not using actual counterfactual token pairs.  Instead in these lines and those that follow, we are calculating the effects of an imagined modification to a token, in which an upstream component’s contribution to the token had been removed.
>
> Turning to lines 759 - 767, here we are describing a method for measuring model performance.  Again, for the IOI task, one can evaluate a model's performance by measuring the logit difference between the correct name (the IO name, "Mary") and the incorrect name (the S name, "John"). These logits are available after a single forward pass, and comparing them serves as a success metric. In all of our tasks, we use a success metric based on the logit difference between “success tokens” and “failure tokens.” For the GP task, the success token is either “he” or “she,” while the corresponding failure token is the other pronoun. Our use of these metrics is consistent with the extensive literature on these tasks.
>
> Lines 768 - 771 are connecting the success metric with a “success direction” in activation space at the model output.   For any success metric there is a corresponding direction, akin to an initial “signal”, that can be used to start our tracing method.
>
> That said, we note that our method does not fundamentally rely on this evaluation scheme. We can trace any linear combination of the unembedding vectors, that is, any direction in the logit space, by defining a corresponding “success direction” (line 769). For instance, we could trace the direction corresponding to the top-k tokens for any given prompt.
>
> We appreciate the opportunity to tease out these different threads, and these questions have been helpful in exposing the need for clearer language around counterfactuals, which we will add to the paper.
> - Thank you for making this important connection. The concept of "binding" is indeed relevant to our work. We view the challenge of correctly associating attributes with entities as a crucial application of the more general attention-causal mechanisms we study. As the reviewer correctly notes, the four works cited identify specific low-rank subspaces that control these binding functions. We hypothesize that these "binding subspaces" are the concrete, circuit-level implementation of the causal mechanism our work aims to identify. Therefore, we expect our method not only to find similar low-dimensional directions but also to provide a mechanistic explanation for how attention heads manipulate information within these subspaces to achieve binding. We agree this connection deserves a more detailed discussion, which we will add to the related work section of the paper.
>
> ### Other: Minor Typos
> - Thank you for pointing these out. We will correct them in the revision.

---

> ### Comment · Reviewer_kEJw · 2025-08-05
>
> Thank you for the response, particularly related to explaining your various uses of "counterfactual" and how they differ throughout the paper. It is more clear to me now and hope some of these changes will be incorporated into the final paper.
>
> I have also read through the other reviewers' responses and rebuttals and I think the proposed changes (specifically in discussion with reviewer Zss5) will be helpful for increasing the clarity of the paper. With the rebuttal response addressing my concerns, I am inclined to leave my score as Accept.

---

### Official Review · Reviewer_5oqm · 2025-06-30

**Clarity:** 2
**Significance:** 3
**Originality:** 3
**Rating:** 3
**Confidence:** 2

**Summary:**

This paper investigates the internal mechanisms of transformer-based language models by focusing on how low-dimensional features influence attention behavior. In particular, this paper examines how such representations function as attention-causal communication channels written into and read from token positions that have a provable causal influence on attention patterns. The authors then introduce a method that jointly analyzes model weights and representations (without relying on counterfactual interventions) to isolate these attention-causal signals. This enables prompt-specific circuit discovery and sheds light on latent mechanisms in attention.

**Questions:**

1. Line 201: feature set ${f^i}$, ${g^i}$: what are $f$, $g$ and $i$?  Please define these formally.

2. What is the x-axis of Figure 2? Could you help parse the plot?

3. Similarly can you help parse Figure 6 first row? What is x-axis for the dendrogram?

3. How does attention-causal communication relate to or improve upon prior tools? Are there cases where prior methods fail but ACC succeeds?



**Typos and Wording Suggestions**

Line 48:  "Our attack on this question" & Line 138: "We first attack the problem": "approach to"& "address" or "tackle" is more appropriate.

Line 179: “than” should be “then.”

**Ethical Concerns:**

["NO or VERY MINOR ethics concerns only"]

**Final Justification:**

The authors clarified some of my questions regarding notations, which helped address some of my initial confusion. However, key concerns remain unresolved. Most importantly, the current form of the paper is not ready for publication: the notation system is messy, and the figures are difficult to interpret. While some of the results are conceptually interesting, the authors themselves acknowledge a lack of intuitive/theoretically grounded explanation, which limits the accessibility/rigor of the work. In addition, the absence of comparisons with relevant methods reduces the empirical soundness. Overall, while I see potential in the direction, the clarity, rigor, and completeness of the presentation need substantial improvement. Thus I am keeping my score and recommend rejection.

**Limitations:**

Yes

**Quality:**

2

**Strengths And Weaknesses:**

**Strengths**
1.  The paper studies how and where task-relevant signals reside in LLMs. The idea of sparse decomposition to uncover key subspaces is conceptually compelling.

2. The proposed attention-based tracing mechanism offers a new lens to analyze the flow of information in transformer blocks without counterfactuals, complementing existing interpretability tools.

3. The paper evaluates the method across a variety of known benchmarks such as IOI and GP, demonstrating the potential generality of the idea.


**Weaknesses**:

1. The notational system, particularly in the sparse decomposition section, is dense and in places undefined or ambiguous. For example, $\mathcal{S}^{\text{lads}}$ (line 212) is introduced in natural language without a formal definition, which impedes understanding.

2. Missing notation for key variables: $f^i$, $g^i$ (line 201) are used without prior introduction or definition, making the formulation difficult to follow.

3. Lack of theoretical foundation: While the empirical motivation is strong, the paper lacks a rigorous theoretical analysis of the proposed decomposition. Key questions remain unanswered: for example, why should attention show sparse decomposition? (as proposed by the authors). Although lines 185–205 attempt to provide some intuition, the discussion is informal and primarily heuristic. Lemma 2 is a standard result in linear algebra and does not offer novel insight specific to the proposed decomposition. As a result, the interpretability claims remain largely empirical and would benefit from stronger theoretical support.

4. The probing tasks (IOI, GT, GP) are referenced but not described. A brief summary of what these tasks entail would be helpful for readers unfamiliar with them.

5. While the paper includes performance comparisons with other circuit identification methods, it lacks a clear conceptual and methodological comparison with widely used interpretability approaches such as attention rollout. This makes it difficult to assess the unique value and practical advantages of the proposed framework. Additionally, the method itself is quite complex (the calculation in Section 4). A more streamlined or intuitive explanation (especially from a practitioner’s perspective) would enhance accessibility and broaden the potential impact of the work.

---

> ### Author Rebuttal · Authors · 2025-07-30
>
> We thank the reviewer for their helpful review of our paper.
>
> ### Weaknesses:
> 1. Thanks for pointing this out.   We’ll provide a clearer definition of $S^{\ell ads}$ in the revision.   Specifically, each term in (3) is the amount that one singular vector adds to the relative attention $c^{\ell ads}$.  We take the terms from (3) and sort them in decreasing order, and find the shortest initial sequence of this list that sums to $c^{\ell ads}$.  Because most terms in (3) are quite small due to sparse decomposition, our method gives a small number of singular vectors.   The indices of these singular vectors are $S^{\ell ads}$.
> 2. Thank you for pointing out the lack of explanation of notation.   We will add an explanation of this notation to the paper in the location you pointed out.   Briefly, the sets {$f^i$}, {$g^i$} are intended to represent sets of features present in a token.   Each feature, eg, $f^0$, is assumed to be a single direction in activation space, following the linear representation hypothesis.   We go into more detail in our next answer immediately below.
> 3. We appreciate the reviewer’s interest in theoretical justification for sparse decomposition of attention scores, which we share.   We recognize that, due to space limitations, we were not able to provide a more detailed explanation of why attention scores should show sparse decomposition.
>
> We start from the linear representation hypothesis, which holds that important features are often encoded in one-dimensional or low-dimensional subspaces. We provide a number of cites in the introduction to papers that demonstrate this phenomenon for an enormous variety of different kinds of features [12, 19-29].   Consider a specific feature, such as the fact that a token corresponds to a person’s name.   This feature may be represented by a 1-D direction in activation space.  We denote such a direction $f^0$.   Hence we can think of the model activation carrying this feature as $x = b + f^0$.
>
> Consider an attention head, one of whose roles is to attend to the personal-name feature.  It calculates its attention score as $x^\top \Omega y$. Then the process of model training will tune $\Omega$ and $f^0$ so that a left singular vector of $\Omega$ will tend to be in the same direction as $f^0$.   This is the significance of Lemma 2.   Note that the other singular vectors cannot lie in the same direction as $f^0$, because singular vectors form an orthogonal set.   Hence, if the personal-name feature were the only feature that the head attended to, then decomposing the attention score $x^\top \Omega y$ in the singular vectors of $\Omega$ would yield a sparse decomposition.  That is, the one singular vector that is in the direction of $f^0$ would yield a large contribution to the attention score, and the other directions would not contribute significantly.
>
> To understand how this extends to the situation where there are multiple features that the head attends to, we note that previous work [19] has shown that features that co-occur will naturally tend to be represented by orthogonal or nearly-orthogonal directions.   That is, features such as “personal name”, “place name”, “organization name” will tend to be represented by vectors $f^0, f^1, f^2$ which are mutually orthogonal or nearly so.   Then by the same argument, model training will tend to align different left singular vectors of $\Omega$ with each of $f^0, f^1, f^2$, again leading to a sparse decomposition of attention when *one* of those features is present in the input $x$.
>
> Finally, in the general case where more features are important to the attention head than there are available singular vectors (dimensions $R$), the tendency will be for features to be approximately orthogonal as explained in [19].   In that case, each feature will tend to align with a small number of singular vectors of $\Omega$, still leading to a sparse decomposition of attention when one of those features is detected.
>
> The above is not a formal proof, which would involve defining a distribution over features and a model of the training process, and would be outside the scope of our paper.  Rather it is an explanation of why sparse attention decomposition can arise.  A formal proof, along with experiments that demonstrate the emergence of sparse decomposition during the training process, is a subject of our current ongoing work.
>
> Moreover, we note that the authors in [58] make assumptions that essentially follow the reasoning we present above, although they do not lay out the reasoning in detail as we do here.
>
> 4. Indeed, these descriptions are missing from the body of the paper. The tasks were described in Appendix J due to the lack of space, but we will add short descriptions of the tasks to the body of the paper to aid the reader.
>
> 5. Thank you for pointing this out. In attention rollout, the identities of input tokens are linearly combined through the layers based on the attention weights. Thus attention rollout is a heuristic method to approximately track information flow through the model.  It does not attempt to identify causal communication. Rather, being a heuristic, it is typically used as a tool to make general observations about what tokens a model is attending to. In contrast, our method identifies the specific low-dimensional components of a token that are responsible for the fact that a head attends to the token. Accordingly, our method provides a more precise and reliable understanding of why an attention head attends to a particular token pair.
>
> We agree that a more complete methodological comparison with attention rollout and other internal tracing methods would clarify our contributions. We also agree that having a more intuitive explanation of the entire methodology would make the paper more accessible and easier to understand. As discussed in our response to Reviewer 1, we believe we can use space currently allocated to Figure 1 to address these needs.
>
> ### Questions:
>
> 1. We address this question in answer #2 (and #3) above.
> 2. Using the definition of $S^{\ell ads}$ that we gave in answer #1 above, the x-axis of Figure 2 is $\frac{|S^{\ell ads}|}{R}$, where $R$ is the dimensionality of the attention heads (line 93) and consequently the rank of the $\Omega^{\ell a}$ matrix. Therefore, the point of Figure 2 is that, for multiple models and tasks, the attention weights are sparsely decomposable in the singular vectors, almost always using a very small number of singular vectors.
> 3. The x-axis of the dendrogram represents all the control signals (D-dimensional vectors) used in a forward pass that were identified by our selection process (lines 332-333). The colors are used to represent the cluster that the control signal falls in, where clustering is based on cosine similarity.
> 4. We appreciate the opportunity to clarify how using attention-causal communication improves over previous methods.
> With respect to circuit tracing, the dominant strategies used for circuit tracing are activation patching or path-patching [7, 8, 11, 42-44].   These methods are still extremely time consuming and require the careful and subtle construction of counterfactuals to expose circuits.   Each time one seeks a circuit for a new task, one must carefully design counterfactual task instances and then run many forward passes of the model, moving activations from the “corrupted” execution to the “clean” execution. Importantly, there are cases where activation patching can fail - instances where downstream behaviors by the model “undo” the effects of the patching.
>
> Our method does not use activation patching, and so avoids problems in which downstream components “undo” the effects of the patching.   Further, it allows for tracing without counterfactuals; the only prompt needed is the one that is to be traced.   Finally, our method performs the trace in a single forward pass – which is much more efficient.
>
> As we discuss in our response to Reviewer Zss5 above, our paper also improves on the tracing method in [51].    Briefly, the method in [51] does not work for models in general – it only worked in [51] for the single case of GPT-2.   In fact, we have found that the method in [51] does not work for Pythia. Our method is based on a formal problem that we solve in a provable way, and thus works on any model (we show its success on GPT-2, Pythia, and Gemma-2).
>
> ### Typos and Wording Suggestions:
> - Thank you for pointing these out. We will correct them in the revision.

---

> ### Comment · Reviewer_5oqm · 2025-08-06
>
> Thank you for the detailed rebuttal, which helped clarify some of my initial confusion. After reviewing the response and other reviews, I agree with Reviewer Zss5 that substantial revisions are needed both in rewriting the text and improving the figures. In its current form, the paper is not ready for publication. In particular, the lack of intuitive/theoretically grounded explanation limits the accessibility/rigor of the work. In addition, the absence of comparisons with relevant methods reduces the empirical soundness. While I see potential in the direction, the clarity, rigor, and completeness of the presentation need substantial improvement. I will keep my score and encourage the authors to improve the clarity and soundness of the work based on the feedback provided.

---

> > ### Author Response · Authors · 2025-08-08
> >
> > We have tried to address the reviewer’s concerns, and hope that since Reviewer Zss5 feels that our revisions are acceptable, the reviewer will review them as well as our rebuttal and also consider whether their concerns have been addressed.

---

### Official Review · Reviewer_Zss5 · 2025-07-03

**Clarity:** 2
**Significance:** 2
**Originality:** 2
**Rating:** 4
**Confidence:** 5

**Summary:**

This paper looks at the the low-rank structure present in the query-key computation of language models. The authors find that singular vectors encode signals across the layers that communicate and can be found automatically (see prior work)

**Questions:**

N/A

**Ethical Concerns:**

["NO or VERY MINOR ethics concerns only"]

**Final Justification:**

Thank you to the authors for engaging so much in the discussion and considering my points. After discussing with the authors, I'm deciding to raise my score. The main reasons are that the authors helped me understand the significance of their work more clearly, and I believe we have consensus on reframing these contributions more fairly to prior work. I do think the required revisions to the writing are significant, but are also doable in time before the camera ready deadline, and I trust that the revisions can be made. Thank you for the hard work in the short discussion period

**Quality:**

3

**Strengths And Weaknesses:**

# Strengths
* The proposed research direction is interesting and important. The authors do an adequate job of formalizing and exploring the ideas of attention-causal structure.
* The proposed solution for materializing QK with models that use relative positional embeddings (RoPE) is a nice contribution and could be widely applicable to other interpretability work
* The paper is thorough and provides a lot of detail in the appendices
* The extension of the singular vector method analysis is to the GT and GP tasks are somewhat interesting, but are the expectation given prior work.
* The authors choose good baselines for circuit comparisons

# Weaknesses
* Lots of jargon and formalisms mask that the paper essentially is a rehash of several prior works. While I think there are some interesting extensions on prior work [50, 51, 58], this paper provides some clarifying formalisms only to replicate **known findings**.

## Known Findings:
* Singular vectors of attention weight matrices encode distinct meaningful signals across layers [50, 51, 58]
* Specifically, the IOI circuit is solved by communication across singular vectors in attention heads [50, 51]
* [51] expands on the ideas of prior work by providing a data dependent measure for selecting singular vectors. This work provides their own version of this, but does not demonstrate why we should prefer there's and offers no novel insight
* The "signals in the stream" analogy is just a rewording of findings from [18] and by extension, the analogy of low rank communication channels in [50]
* Section 4: "Tracing circuits" is not a novel contribution and the language around circuit tracing is very similar to the "circuit tracing" and results described in [51] and does not expand on it meaningfully
* Distinct semantic feature matching performed by singular vectors of [58 <- vision models]. A lot of the analysis resembles [58]


**Overall, I can not recommend this paper for acceptance because of the striking similarity to multiple prior works**


## other weaknesses
* Figure 1 is basically incomprehensible. I don't think it is valuable as an illustration for the method. The colored bars are unclear and there is too much jargon frontloaded. I believe this figure should be remade to better use the space

---

> ### Author Rebuttal · Authors · 2025-07-30
>
> We thank the reviewer for their careful read of our paper and the helpful comments. We appreciate the opportunity to clarify the points raised.
>
> First we’ll address the main thrust of the reviewer’s concerns and then we’ll address the detailed points individually.
> The reviewer is concerned that our paper does not break new ground over [50, 51, and 58]. The closest paper to ours is [51], so we’ll start there.
>
> As the reviewer notes, [51] provides a data dependent measure for selecting singular vectors, which then exposes meaningful communication. The strategy taken in [51] is to select a set of singular vectors that contribute an arbitrary amount (70% of total) to **attention scores** (ie, pre-softmax scores – not attention itself).
>
> First of all, the signals found in [51] have no provable effect on attention itself. Nothing can be said about how much of a head’s attention is actually caused by these signals. That is, there is no formal basis for, nor is there any provable result from the signal identification method used in [51]. As a result, there is no assurance about the nature of the set of signals found in [51] – there may be both false negatives and false positives with respect to attention causality, for example.
>
> In contrast, our paper starts with a formal problem definition of *attention-causal communication* (a concept that is a novel contribution of our paper) and then presents an algorithm that **provably** identifies attention-causal communication. This conceptual advance is important, because it provides a formal definition for what researchers seek in understanding why an attention head attends to a particular token pair. See, for example, the recent blog post “Progress on Attention”  in which prominent researchers point to this question as a key open problem in LLM interpretability (“Our biggest remaining question is how to understand *why models attend where they do*.”).
>
> Second, and related to this flaw, the method of [51] **does not work in general**. For example, it cannot be used on Pythia. We have worked with the method of [51] from the authors’ code and tried to use it on Pythia. We find that the key idea of [51] is based on assumptions that do not hold in general, and do not hold in Pythia specifically. This is perhaps not surprising for an approach with no formal justification. To explain, the key problem is that many attention scores in Pythia are actually *negative*. There is no way to apply the approach of [51] to situations where the attention scores are negative. The method of [51] fundamentally assumes that moving the attention score of a token pair towards zero will decrease the attention on the pair – but this is *not true* for negative attention scores.
>
> Our development of relative attention as the proper measure for selecting singular vectors is therefore a critical advance over [51], and part of the reason it was possible for us to apply our method to all three of GPT-2, Pythia, and Gemma-2 (whereas [51] only worked with GPT-2). Furthermore, the decision to define and use relative attention is not obvious; it’s important that relative attention (a linear function) has a provable relationship with attention (a nonlinear function) as shown in Lemma 1. This fact is key to being able to attribute attention causality for a particular head to components upstream of that head, which relies on the linearity of relative attention.
>
> The reviewer states that our paper “does not demonstrate why we should prefer” our paper over [51]. This is a fair point. Thanks to the reviewer’s comments we realize we should have made these points clearly in the manuscript and we will definitely make these points clearly in the revision.
>
> Turning to paper [50], indeed that paper shows the existence of low-rank communication channels between attention heads. It is important to observe that [50] only provides one concrete example for the IOI task. This example is in GPT-2 and is a particular communication from four inhibition heads to one name mover head. Importantly, as the paper admits, the authors were looking for this channel because they knew from prior work that the inhibition heads communicate with the name mover. This example underscores the fact that the method used by the authors in [50] is entirely **manual and heuristic**, based on examining plots involving pre-selected pairs of heads. The point is that paper [50] **does not provide**, or attempt to provide, **a general or automatic method** for identifying all the important communication occurring in the IOI task, nor does it attempt to provide a general method for identifying which attention heads are communicating. In contrast, our paper presents a systematic, general method, with provable results, for finding all the attention-causal communication present, and identifying all the heads that are contributing, and it does so without assuming that circuits have been traced in advance (unlike [50]). We mention here again that the analysis of the IOI task in GPT-2 does not generalize to other models without making use of significant new ideas (such as relative attention) that we contribute.
>
> Next, regarding paper [58], indeed that paper focuses on semantic feature matching in vision transformers, performed using singular vectors. The main goal in [58] is to show that vision transformers do not just attend to similar objects, but also (in later layers) perform contextual analysis. As such, we believe that [58] is **complementary** to our work. The authors in [58] are focused on exploring the meaning of visual features that correspond to singular vectors. Our work does not dig into the semantics of features but instead focuses on systematically extracting all the relevant features (signals) that are causal for attention. On the other hand, the fact that [58] shows that so many features found using singular vectors of QK matrices are interpretable **adds motivation** to our method, since we show how to extract those (attention-causal) features systematically in a manner that is robust across models and tasks.
>
> Regarding the bullet points:
> - *Lots of jargon and formalisms mask that the paper essentially is a rehash of several prior works. While I think there are some interesting extensions on prior work [50, 51, 58], this paper provides some clarifying formalisms only to replicate **known findings**.*
> We appreciate the opportunity to clarify the ways in which our work advances the field. We don’t claim that we are the first to observe the importance of singular vectors of QK matrices as tools for interpretability. Indeed, we cite [50, 51, 58] as important prior work showing that attention heads communicate in low-dimensional channels identifiable using singular vectors of QK matrices. The focus of our contribution in Section 4 is on showing how to turn this observation into a formalized tracing strategy that is more efficient and easier to use than prior methods (eg, more efficient than using counterfactuals and patching) and that has provable properties and is robust across model architectures (unlike [51]).
> - *Singular vectors of attention weight matrices encode distinct meaningful signals across layers [50, 51, 58]*
> Indeed, as discussed above, this fact is an important background for our paper; we don’t claim this as a contribution of our paper.
> - *Specifically, the IOI circuit is solved by communication across singular vectors in attention heads [50, 51]*
> While both [50] and [51] look at the IOI task via singular vectors, neither offer a robust and systematic solution, as explained above. The other tasks we study (GP, GT) have not been addressed before, and showing that the singular vector method works well for those as well is important evidence for the robustness of the method.
> - *[51] expands on the ideas of prior work by providing a data dependent measure for selecting singular vectors. This work provides their own version of this, but does not demonstrate why we should prefer there's and offers no novel insight*
> As discussed above, we understand now that we did not explain in our submission why our method is a distinct advance over [51]. We explain that above, and plan to clearly articulate our advance over [51] in the revision.
> - *The "signals in the stream" analogy is just a rewording of findings from [18] and by extension, the analogy of low rank communication channels in [50]*
> Understood. Removing it from the title would focus the title better on our paper’s contribution, which we will do in the revision.
> - *Section 4: "Tracing circuits" is not a novel contribution and the language around circuit tracing is very similar to the "circuit tracing" and results described in [51] and does not expand on it meaningfully*
> While much prior work has focused on tracing circuits, the dominant strategies are still extremely time consuming and require the careful and subtle construction of counterfactuals to expose circuits. Tracing without counterfactuals, and in a single forward pass, offers the field a much more efficient tool.
> As described above, our work goes beyond [51] by providing a formal problem definition based on the new concept of attention-causal communication, presenting a provable and efficient solution to that problem, and developing new concepts (relative attention) needed for working with models beyond GPT-2.
> - *Distinct semantic feature matching performed by singular vectors of [58 <- vision models]. A lot of the analysis resembles [58]*
> As described above, as [58] focuses on interpreting the features found via singular vector analysis of QK matrices, we see it as complementary to our work, and providing motivation for the approach we take.
> - Regarding Figure 1, we appreciate the reviewer’s frank appraisal that this figure doesn’t work. This is helpful criticism, and we can see in that light that Figure 1 should be removed and the space used more profitably.

---

> > ### Comment · Reviewer_Zss5 · 2025-08-04
> >
> > Thank you for the very detailed response. I have read and understand the points being made, and I think a lot of good points are raised, but I still take issue with the framing of the paper.
> >
> > > Second, and related to this flaw, the method of [51] does not work in general. For example, it cannot be used on Pythia. We have worked with the method of [51] from the authors’ code and tried to use it on Pythia. We find that the key idea of [51] is based on assumptions that do not hold in general, and do not hold in Pythia specifically. This is perhaps not surprising for an approach with no formal justification
> >
> > This is helpful context, and something I might have not understood (in particular the negative attention scores is a nice explanation). The authors' response as to why this should be preferred over [51] is much more clear to me.
> >
> > > It is important to observe that [50] only provides one concrete example for the IOI task. This example is in GPT-2 and is a particular communication from four inhibition heads to one name mover head
> >
> > Correct me if I'm wrong, but **this statement is not true**. [50] includes results for a duplicate token -> inhibition head circuit for GPT-2 as well as inhibition head OV singular vector experiments on Pythia (in the appendix). They also performed novel circuit discovery on a new head without counterfactuals in the Appendix. In fact, they did so based on connections in between head weights without any data, but I find that their process is much more manual than what is proposed here, and agree that the current presented work is more generalizable.
> >
> > The main issue I think I have is the writing and positioning of the contribution. I would be willing and motivated to raise my score if this was given a rewrite, as I don't think it's productive to publish it as is. I think the current most interesting contributions are providing a more rigorous and principled improvement to finding such channels. I am actually very interested in seeing work like that published, but the writing is confusing. Take for example, this line in the abstract:
> >
> > > Our methods do not make use of counterfactual inputs to the model, but rather work by joint analysis of model weights and model representations to isolate signals – attention-causal features
> >
> > Is extremely similar to lines found in [51] and also highly related to the communication channels described in [50]. You _can_ just frame this as a followup study of these two works, which I think it is. I don't think that inhibits the novelty of the work, and in its current state, I find that burying the lead of similarity to prior work is a little problematic. Other issues, e.g., the confusing Figure 1 without much explanation, multiple use of the term "counterfactual" (raised by reviewer kEJw) make the contribution unclear in a different way.
> >
> > I will leave as a summary comment: **This work is a clear improvement over the less scalable/automated approaches of previous work for analyzing singular vector communication, but I find the framing of its contributions confusing, as it positions the contributions around sparse signals in the residual stream as more novel than they are [50, 51, 58] (and e.g., the title "Sparse signals in the stream"). Overall, I think a lot of text needs to be rewritten and figures redone, and this might be a significant rewrite. Therefore, I believe this may not be ready for publication, but think that the paper does contribute meaningfully to the prior literature.**

---

> > > ### Author Response · Authors · 2025-08-04
> > >
> > > We understand the reviewer’s observations about the framing, especially with respect to [50, 51]. We agree that it would be more appropriate to reframe the paper as a follow-up study of those two works. We will work on drafting changes over the next 24 hours and will respond with proposed edits tomorrow, to include the suggestions made by the reviewer.
> > >
> > > Regarding other points: we accept the reviewer’s judgment that Figure 1 isn’t helpful, and we will remove it and use the space to improve the paper, including more clearly discussing the role of the two different kinds of counterfactuals.

---

> > > ### Author Response · Authors · 2025-08-05
> > >
> > > We appreciate the referee’s observations about the framing, especially with respect to [50, 51]. We agree that it would be more appropriate to reframe the paper as a follow-up study of those two works.  Below, we have tried to do that and we hope that the referee sees our paper as worthy of acceptance given that framing.
> > >
> > > We propose to adjust the framing of the paper in seven places: title, abstract, introduction, related work, Section 2, Section 3, and conclusion.
> > >
> > > ### 1. Title
> > >
> > > Change to “Pinpointing Attention-Causal Communication in Language Models” (removed “Signals in the Stream”)
> > >
> > > ### 2. Abstract:
> > >
> > > **Remove** this passage (part of which was pointed to by the reviewer): lines 9-14:
> > > *Our methods do not make use of counterfactual inputs to the model, but rather work by joint analysis of model weights and model representations to isolate signals -- attention-causal features.  These signals serve as communication channels from upstream model components to the attention head.  We show that by identifying the signals present when a model processes a prompt, we can perform prompt-specific circuit discovery without counterfactuals.*
> > >
> > > **Replace** the above with this passage:
> > > *The starting point for our method is prior work [50, 51, 58] showing that model components make use of low dimensional communication channels that can be exposed by the singular vectors of QK matrices. Our contribution is to provide a rigorous and principled approach to finding those channels and isolating the signals they contain. We show that by identifying those signals, we can perform prompt-specific circuit discovery in a single forward pass.*
> > >
> > > ### 3. Introduction
> > >
> > > **Remove** this passage:  **lines 48-50**:
> > > *Our attack on this question starts from analysis of model weights, specifically the QK matrices of attention heads. We point out that a natural consequence of the Courant-Fischer theorem is that the singular vectors of QK matrices will tend to be aligned with important features in residuals.*
> > >
> > > **Replace** the above with this passage:
> > > *Our attack on this question starts from prior work [50, 51, 58] showing that the singular vectors of QK matrices will tend to be aligned with important features in residuals.*
> > >
> > > ###  4. Related work:
> > >
> > > - **Delete** this passage (ie, no replacement): **lines 74-75:**
> > > *Our work adds a new direction by showing that QK matrices can be analyzed to expose the low dimensionality of features.*
> > >
> > > - **Remove** this passage: **lines 88-90**:
> > > *Our approach to signal identification is consistent with, and extends, recent work showing that model components often communicate in low dimensional subspaces defined by the spectral decomposition of QK matrices [50, 51, 58].*
> > >
> > >     **Replace** the above with this passage:
> > >     *As stated above, our approach to signal identification builds on prior work showing that model components often communicate in low dimensional subspaces defined by the spectral decomposition of QK matrices [50, 51, 58].*
> > >
> > > ### 5. Section 2
> > >
> > > - **Remove** this passage: **lines 169-170**:
> > > *Then the following hypothesis drives our approach:*
> > >
> > >     **Replace** the above with this passage:
> > >     *Then the following hypothesis (adapted from [51]) drives our approach:*
> > >
> > > - **Remove** this passage: **line 172**:
> > > *Hypothesis (Sparse Attention Decomposition)*
> > >
> > >     **Replace** the above with this passage:
> > >     *Hypothesis (Sparse Attention Decomposition) [51]*
> > >
> > > - **Remove** this passage: **lines 182-184**:
> > > *This strategy is broadly consistent with a number of recent studies that have shown that the singular vectors of QK matrices provide useful decompositions of attention and inter-layer communication [50, 51, 58].*
> > >
> > >     **Replace** the above with this passage:
> > >     *This strategy is based on prior work [50, 51, 58] showing that the singular vectors of QK matrices provide useful decompositions of attention and inter-layer communication.*
> > >
> > > ### 6. Section 3
> > >
> > > **Remove** this passage: **line 224**:
> > > *Next we show that the phenomenon of sparse attention decomposition is ubiquitous.*
> > >
> > > **Replace** the above with this passage:
> > > *Next we confirm prior work showing that the phenomenon of sparse attention decomposition is ubiquitous [50, 51, 58].*
> > >
> > > ### 7. Conclusion
> > >
> > > **Remove** this passage: **line 369**:
> > > *We observe that attention is generally sparsely decomposable in the QK basis,*
> > >
> > > **Replace** the above with this passage:
> > > *We build on previous observations that attention is generally sparsely decomposable in the QK basis [50, 51, 58]*

---

> > > > ### Comment · Reviewer_Zss5 · 2025-08-08
> > > >
> > > > Thank you for the hard work from the authors in the short discussion period. The authors have helped me understand their contributions more clearly, and I think we have consensus on the revisions that need to be made to the writing to make it camera ready. The revisions are definitely significant, covering lot of the body of the paper and a few figures, but personally I think it's reasonable to do in the time before the camera ready deadline, and only cover cosmetic/framing (not experiments), so I am raising my score. Thank you for such detailed revisions

---

> > > > > ### Author Response · Authors · 2025-08-08
> > > > >
> > > > > We are very appreciative of the reviewer's contributions, which have definitely improved the paper. We appreciate the reviewer’s willingness to consider our feedback and the care with which they read the paper.

---

### Note · Authors · 2025-08-11

- The review process has been constructive; our paper is significantly improved due to reviewer feedback. We have addressed all points raised in detail.

- Reviewers tbcy and kEJw have been **in favor of acceptance** and we appreciate their suggested improvements.

- Reviewer Zss5 initially raised two concerns:
  1. Contribution over prior work. We cited three pieces of prior work that substantiate a key idea (namely, that attention heads communicate in low-dimensional channels exposed by the SVD of QK matrices). Reviewer Zss5 questioned whether our work went beyond those papers, in particular papers [50] and [51].  We explained in detail why and how our paper goes beyond those works in significant ways, and acknowledged the need to include that explanation in the revision. **Reviewer Zss5 indicated that our explanation made our contribution clear.**
  2. Framing. Reviewer Zss5 felt our paper was not properly contextualized. We agreed and **proposed detailed line-by-line changes** to properly frame our work. **Reviewer Zss5 indicated they agreed with this new framing.**

  This productive interaction has distinctly improved the paper, and our understanding is that Zss5 now supports acceptance.

- Reviewer 5oqm initially rated the paper a Weak Reject with low confidence. We provided a detailed point-by-point response, but 5oqm did not engage with it. Their final comments ask for:
  - “intuitive/theoretically grounded explanation”. We note that we provide an **intuitive explanation both in the paper** (lines 185-205) and **in more detail in our rebuttal** (which we propose to incorporate in the final version).
  - “Comparisons with relevant methods”. We note that we provide **comparison with key alternative methods** of tracing in the paper: we explain methodological advantages on lines 82-87, and performance comparisons on lines 304-317, Fig. 5, and App K. Moreover, the revisions agreed upon with Reviewer Zss5 will drastically improve the conceptual comparison to prior work.

  We want to ensure 5oqm has the latest context for our paper. Reviewer 5oqm’s final comments were posted while our discussion with Zss5 was still ongoing. Since that time, Zss5 has confirmed that their initial concerns are now addressed and has raised their score. Given that Reviewer 5oqm’s assessment was partly based on the perceived concerns of Reviewer Zss5, we would like to ask whether Reviewer 5oqm would reconsider whether the rebuttal we provided addresses each of their points.

---

### Decision · Program_Chairs · 2025-09-17

**Decision:**

Accept (poster)

**Comment:**

The authors decompose (query-value modules in) attention mechanism components into low-rank structures to identify features that are causally related to attention patterns. All reviewers agreed that this paper offers several nice contributions which build naturally on prior work in interpretability.

I do think the authors should take care to address the issue of contextualizing the current contribution with respect to prior work (per the discussion with reviewer Zss5), and to ensure the notation is clear and consistent throughout (5oqm).